# Autophagy-mediated apoptosis eliminates aneuploid cells in a mouse model of chromosome mosaicism

Shruti Singla[1], Lisa K. Iwamoto-Stohl[1], Meng Zhu [1] & Magdalena Zernicka-Goetz [1,2]✉

The high incidence of aneuploidy in the embryo is considered the principal cause for low human fecundity. However, the prevalence of aneuploidy dramatically declines as pregnancy progresses, with the steepest drop occurring as the embryo completes implantation. Despite the fact that the plasticity of the embryo in dealing with aneuploidy is fundamental to normal development, the mechanisms responsible for eliminating aneuploid cells are unclear. Here, using a mouse model of chromosome mosaicism, we show that aneuploid cells are preferentially eliminated from the embryonic lineage in a p53-dependent process involving both autophagy and apoptosis before, during and after implantation. Moreover, we show that diploid cells in mosaic embryos undertake compensatory proliferation during the implantation stages to confer embryonic viability. Together, our results indicate a close link between aneuploidy, autophagy, and apoptosis to refine the embryonic cell population and ensure only chromosomally fit cells proceed through development of the fetus.

[1] Department of Physiology, Development and Neuroscience, University of Cambridge, Downing Street, Cambridge CB2 3EG, UK. [2] Division of Biology and Biological Engineering, California Institute of Technology, 1200 E. California Boulevard, Pasadena, CA 91125, USA. ✉email: magdaz@caltech.edu

Humans exhibit relatively suboptimal fertility compared to other mammals[1]. Studies of early pregnancy loss indicate that only 30% of conceptions progress to live birth[2]. Cytogenetic studies of spontaneous abortions and pre-implantation in vitro fertilised (IVF) embryos show that most of these losses are associated with aneuploidy[2–4]. Interestingly, both normal fertilisation and IVF often give rise to embryos that are mosaics, i.e. contain cells with different chromosome constitutions[5–7]. Although ~60% of pre-implantation IVF embryos exhibit diploid–aneuploid mosaicism[5], our understanding of the embryo's ability to cope with such abnormalities is very limited. The incidence of aneuploidy declines at later stages of development, from 73% in pre-implantation stages to only 0.6% in live births[5,8]. This reduced frequency of aneuploid cells during embryonic development could arise via the preferential allocation of abnormal cells to extra-embryonic lineages[9,10], the self-correction of abnormalities[11] or clonal depletion[12,13]. However, there is insufficient evidence to support any of these processes in human embryo development.

We recently established a mouse model of chromosome mosaicism[14] by acutely inhibiting the spindle assembly checkpoint (SAC)[15]. This model allowed us to discover that aneuploid cells induce different responses in different lineages as the diploid–aneuploid mosaic embryo develops into the blastocyst[14]: apoptosis in the inner cell mass (ICM), which will generate the embryonic lineage and cell cycle delay in the trophectoderm (TE), which will form the placenta. However, we also found that more than half of aneuploid cells still remain by the time of implantation[14]. Whether these aneuploid cells become eliminated after implantation, and if so, when and how this elimination takes place and whether this aneuploid cell elimination affects surrounding diploid cells has remained unknown. We now wish to address these questions.

Here, we examine the fate of aneuploid cells during implantation and early post-implantation development when the cells of the embryonic lineage initiate for the first time extensive proliferation and reorganisation in shape, gene expression, epigenetic signatures and metabolism, laying down the foundation of the developing body. The results we present here demonstrate that aneuploid cells are eliminated from the embryonic lineage by apoptosis, due to proteotoxic stress which in turn activates autophagy, favouring survival of a diploid embryonic cell pool.

## Results

**Aneuploid cell loss in the peri/post-implantation epiblast**. We previously showed that aneuploid cells begin to disappear from the embryonic lineage of the aneuploid mosaic embryos during pre-implantation development, specifically at the blastocyst stage[14]. However, the mechanisms by which aneuploid cells are eliminated and whether aneuploid cells elimination continues as embryos implant have remained unknown. To address these questions, we induced chromosome segregation errors by treating embryos with a well-established small reversible inhibitor, reversine, during four- to eight-cell division to inactivate the SAC[15] and confirmed that this treatment significantly increased the incidence of aneuploidy in comparison to DMSO-treated controls (Supplementary Fig. 1a–d), in agreement with previous results[14]. Thus, we will refer to reversine-treated cells as aneuploid and to control, DMSO-treated cells as diploid throughout for simplicity.

To recognise diploid and aneuploid cells in mosaic embryos, we wished to generate chimeras in which diploid and aneuploid clones could be distinguished from each other. To this end, we treated non-fluorescent embryos with reversine at the four- to eight-cell stage division (aneuploid embryo) and membrane-

targeted red fluorescent (mT/mG) embryos with DMSO (diploid embryo), let embryo develop to the eight-cell stage at which point we isolated individual cells from each other and aggregated four red fluorescent diploid and four non-fluorescent aneuploid cells into the chimeras, and let them develop to the blastocyst stage (Fig. 1a). We also used the same approach to generate control diploid–diploid chimeric embryos.

To specifically monitor the fate of the embryonic lineage beyond the blastocyst stage, we have established an in vitro model in which the extra-embryonic TE is removed by immunosurgery[16] and the isolated ICM is embedded in the Matrigel and an in vitro culture (IVC) implantation medium, which we previously established[17]. We showed that this experimental model allows us to recapitulate the sequential events of implantation and early post-implantation development: formation of the epiblast rosette that undergoes lumenogenesis in its centre to form the pro-amniotic cavity (Supplementary Fig. 2a–c).

In each experiment we generated both aneuploid–diploid and diploid–diploid chimeric embryos and first cultured them to the late blastocyst stage and then removed the TE using immunosurgery, embedded the ICMs in Matrigel and cultured them in IVC medium for 72 h (Fig. 1a). We found that both diploid–diploid and the diploid–aneuploid chimeras developed characteristic post-implantation morphology, in which the primitive endoderm surrounded the epithelial epiblast that formed a lumen in its centre (Fig. 1b). The developmental efficiency of diploid–diploid and diploid–aneuploid chimeras were similar (Supplementary Fig. 3a, b). The relative proportion of non-fluorescent cells was significantly lower in the diploid (red)–aneuploid (non-fluorescent) chimeras than in the diploid (red)–diploid (non-fluorescent) chimeras both in the epiblast and the primitive endoderm (Fig. 1c, Supplementary Fig. 3c, d). Specifically, the frequency of non-fluorescent aneuploid cells in diploid–aneuploid epiblasts was 30.3% while the frequency of non-fluorescent diploid cells in diploid–diploid epiblasts was 64.9% (Fig. 1c). A similar preferential depletion of aneuploid cells was visible in the primitive endoderm, the frequency of non-fluorescent aneuploid cells in diploid–aneuploid primitive endoderm was 38.2%, while the frequency of non-fluorescent diploid cells in diploid–diploid primitive endoderm was 65.9% (Fig. 1c).

Strikingly, nearly half (47.4%) of epiblasts derived from diploid–aneuploid chimeras completely lacked non-fluorescent, aneuploid cells (Supplementary Fig. 3e). In contrast, only 19% of the diploid–diploid chimeras showed a complete loss of the non-fluorescent, diploid cells (Supplementary Fig. 3e). Similarly, 31.6% of diploid–aneuploid chimeras lost the non-fluorescent aneuploid cells in the primitive endoderm, whereas only 10% of the diploid–diploid chimeric primitive endoderm lacked non-fluorescent diploid cells (Supplementary Fig. 3f).

We next wanted to determine the fate of aneuploid cells in the mosaic embryos which were allowed to implant in vivo and in which TE is not removed. To this end, we treated embryos with reversine (or DMSO) at the four- to eight-cell stage transition and generated eight-cell stage diploid–diploid and diploid–aneuploid chimeras, let them develop to the blastocyst stage, at which point we transferred embryos to foster mothers (Fig. 1d). We recovered the chimeric embryos 12 h after implantation and analysed their composition after 36 h of IVC. We found that the relative proportion of non-fluorescent cells was significantly lower in the diploid (red)–aneuploid (non-fluorescent) chimeras than in the diploid (red)–diploid (non-fluorescent) chimeras in the epiblast (Fig. 1e, f, Supplementary Fig. 4a, b). Specifically, the frequency of non-fluorescent aneuploid cells in diploid–aneuploid epiblasts was 19.67% while the frequency of non-fluorescent diploid cells in diploid–diploid epiblasts was 49.99% (Fig. 1f). The frequency of

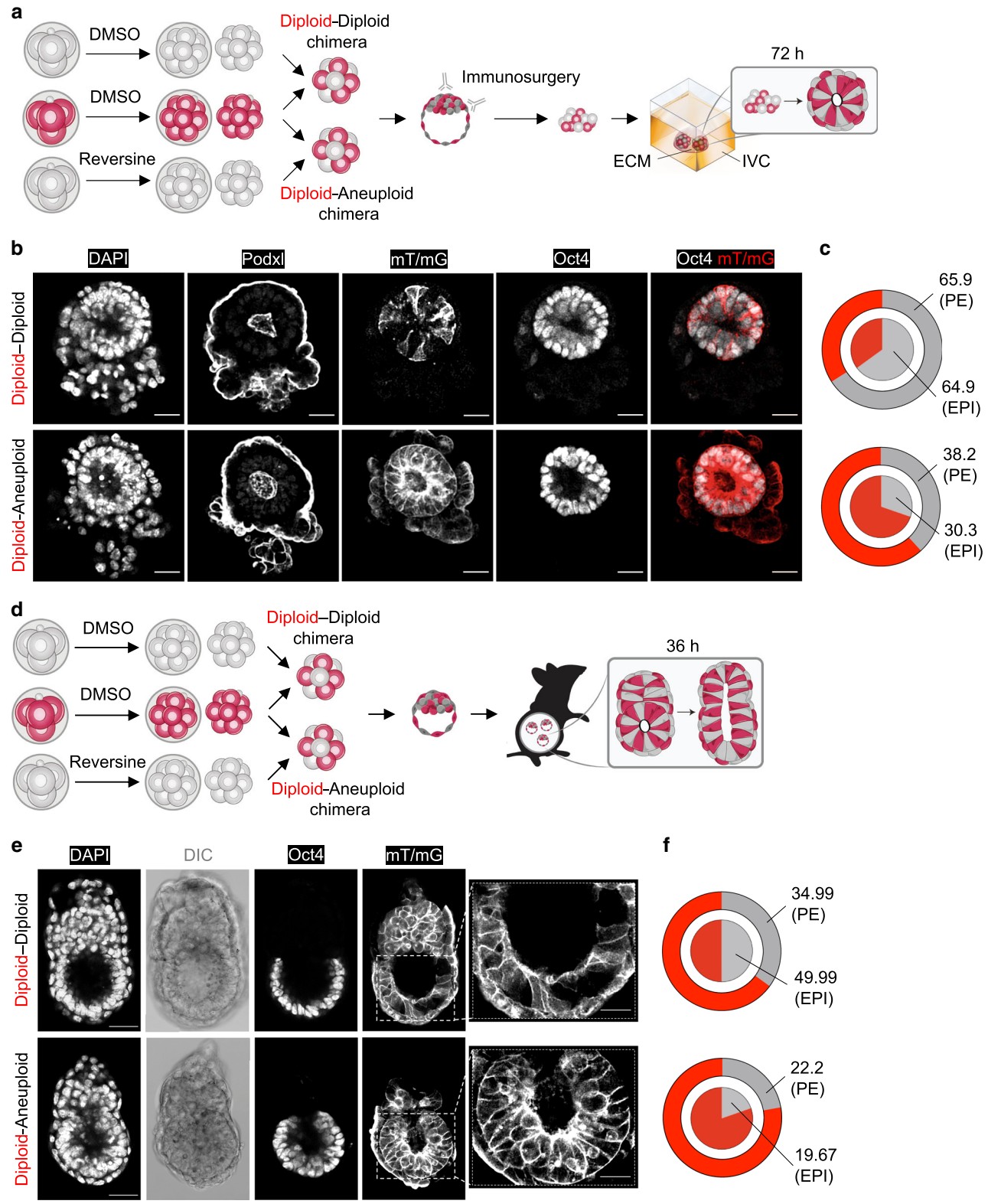

non-fluorescent aneuploid cells in diploid–aneuploid primitive endoderm was 22.2% while the frequency of non-fluorescent diploid cells in diploid–diploid primitive endoderm was 34.99% (Fig. 1f). However, this decrease in the relative proportion of non-fluorescent cells in the diploid (red)–aneuploid (non-fluorescent) chimeras than in the diploid (red)–diploid (non-fluorescent) chimeras was not significant in the primitive endoderm (Supplementary Fig. 4c).

Again, similarly to embryos cultured in vitro, 78.6% of diploid–aneuploid chimeras contained no non-fluorescent aneuploid cells in the epiblast, whereas only 14.3% of the diploid–diploid chimeras lacked non-fluorescent diploid cells in the epiblast (Supplementary Fig. 4d). A similar depletion of aneuploid cells occurred in the primitive endoderm: 62.5% of diploid–aneuploid chimeras contained no non-fluorescent aneuploid cells in the primitive endoderm, whereas 50% of the diploid–diploid chimeras

**Fig. 1 Aneuploid cells become eliminated during peri-implantation epiblast remodelling.** Embryos were treated with reversine (or DMSO) at the four- to eight-cell stage transition and eight-cell chimeras containing a 1:1 ratio of control (diploid) and reversine-treated (aneuploid) cells were constructed from mT/mG (red) diploid cells and non-fluorescent aneuploid cells at the eight-cell stage. **a** At the late blastocyst stage, immunosurgery was performed to isolate the inner cell mass (ICM) from the trophectoderm (TE). Chimeras were embedded in Matrigel and cultured in IVC medium for 72 h to allow development into an epithelised epiblast (EPI) surrounded by a primitive endoderm (PE) layer with a central lumen. **b** In these examples, the diploid–diploid EPI contains both red fluorescent and non-fluorescent cells. Whereas, the majority of the diploid–aneuploid chimera originates from the red fluorescent diploid clone. Scale bars, 30 μm. **c** After culture according to (**a**), average distribution of red fluorescent and non-fluorescent cells was assessed for both types of chimeras in the EPI and PE. For **b** and **c**, diploid–diploid $n = 21$ embryos; $n = 2636$ EPI cells; $n = 3003$ PE cells and diploid–aneuploid $n = 19$ embryos; $n = 1900$ EPI cells; $n = 2306$ PE cells. **d** At the early blastocyst stage, chimeras were transferred to pseudo-pregnant mothers and recovered 12 h after implantation and cultured in IVC medium for 36 h. **e** In these examples, the diploid–diploid EPI contains both red fluorescent and non-fluorescent cells. Whereas, most of the diploid–aneuploid chimera EPI originated from the red fluorescent diploid clone. Scale bars, 40 μm. Squares indicate magnified regions. Scale bars, 20 μm. **f** After culture according to (**d**), average distribution of red fluorescent and non-fluorescent cells was assessed for both types of chimeras in the EPI and PE. For the EPI (**f**), diploid–diploid $n = 7$ embryos; $n = 807$ EPI cells and diploid–aneuploid $n = 14$ embryos; $n = 1640$ EPI cells. For (**e**) and the PE (**f**), diploid–diploid $n = 6$ embryos; $n = 1110$ PE cells and diploid–aneuploid $n = 8$ embryos; $n = 1488$ PE cells. Source data are provided as a Source Data file.

lacked non-fluorescent diploid cells in the primitive endoderm (Supplementary Fig. 4e).

As an alternative approach, we also generated double size diploid–aneuploid mosaic chimeras (16-cell embryos) by aggregating whole 8-cell diploid embryos with whole 8-cell aneuploid embryos and culturing them in IVC medium after immunosurgery to isolate ICM (Supplementary Fig. 5a). In agreement with the above results, we found that both double size diploid–diploid chimeras and diploid–aneuploid chimeras developed the characteristic post-implantation morphology at equivalent rates (Supplementary Fig. 5b, c). These chimeras also showed a significant reduction in the relative proportion of non-fluorescent cells in the diploid (red)–aneuploid (non-fluorescent) chimeras than in the diploid (red)–diploid (non-fluorescent) chimeras both in the epiblast and the primitive endoderm (Supplementary Fig. 5b, d–f). Specifically, the frequency of non-fluorescent aneuploid cells in diploid–aneuploid epiblasts was 33.3%, while the frequency of non-fluorescent diploid cells in diploid–diploid epiblasts was 69% (Supplementary Fig. 5d). The frequency of non-fluorescent aneuploid cells in diploid–aneuploid primitive endoderm was 37.5%, while the frequency of non-fluorescent diploid cells in diploid–diploid primitive endoderm was 74.9% (Supplementary Fig. 5d). Strikingly, 53.3% of the diploid–aneuploid epiblasts contained no non-fluorescent aneuploid cells. In contrast, only 15.4% of the diploid–diploid epiblasts lacked non-fluorescent diploid cells (Supplementary Fig. 5g). Furthermore, 23.3% of diploid–aneuploid chimeras contained no non-fluorescent aneuploid cells in the primitive endoderm, whereas all of the diploid–diploid chimeras contained non-fluorescent cells in the primitive endoderm (Supplementary Fig. 5h). Consistently, in the 16-cell double size chimeras in which the TE was not removed, there was a significant reduction in the frequency of aneuploid cells in the epiblast, but not in the primitive endoderm (Supplementary Fig. 6a–e). Taken together, these results indicate that aneuploid cells are preferentially eliminated during remodelling of the epiblast, both in vitro and in vivo.

**Aneuploid cell depletion from the epiblast by apoptosis.** Since our results indicated a significant elimination of the aneuploid cells from the epiblast, we next decided to focus on this lineage and wished to determine how the aneuploid cells become eliminated from diploid–aneuploid mosaic embryos during epiblast morphogenesis after implantation. To follow the behaviour and survival of aneuploid cells, we generated chimeric embryos in which aneuploid clones were distinguished by their expression of green nuclear (histone H2B-GFP) and diploid cells of red membrane (mT/mG) fluorescent markers, isolated ICMs by

immunosurgery, embedded them in Matrigel and IVC medium and carried out time-lapse imaging under a spinning disc confocal microscope for 72 h (Fig. 2a). Our live embryo imaging revealed several morphological features characteristic of apoptosis[18] in the green fluorescent aneuploid cells. This included nuclear condensation, followed by the formation of the apoptotic bodies and subsequent removal of the cellular debris (Fig. 2a, Supplementary Movie 1). We found that double size diploid–aneuploid chimeras also displayed similar features indicative of apoptosis of the aneuploid cells in the epiblast and engulfment of the apoptotic bodies by the neighbouring red fluorescent control cells (Fig. 2b, Supplementary Movie 2). We also observed apoptosis of the aneuploid cells in the post-implantation epiblasts of diploid–aneuploid chimeras in which the TE was not removed and which developed through implantation in vivo (Fig. 2c, Supplementary Movie 3). These observations suggest that aneuploid cells are preferentially depleted from the epiblast by apoptosis and the apoptotic debris was gradually cleared from the embryo during the peri-implantation stages of development both in vivo and in vitro.

**Size regulation of diploid–aneuploid mosaic epiblasts.** In order to determine the effect of preferential elimination of aneuploid epiblast cells from diploid–aneuploid embryos on the overall mosaic embryo development, we next analysed the mosaic epiblast at the end of peri-implantation development in greater detail. Strikingly, we found that despite the depletion of aneuploid cells from the diploid (red)–aneuploid chimera epiblast, diploid–diploid and diploid–aneuploid epiblasts had a similar average cell number after 72 h of IVC culture (Fig. 3a). Both double size diploid–diploid chimeras and diploid–aneuploid chimeras also contained an equivalent number of cells in the epiblast after 72 h of IVC culture (Fig. 3b). The average number of cells in the early post-implantation epiblast was also similar in diploid–diploid and diploid–aneuploid chimeras (Fig. 3c), as well as in double size chimeras (Supplementary Fig. 6f) in which the TE was not removed and which were allowed to implant in vivo. These results suggested the possibility of compensatory proliferation of diploid cells to regulate the total number of epiblast cells.

To assess cell proliferation in the epiblasts, we generated again mosaic aneuploid–diploid and diploid–diploid chimeras, as above (Fig. 1a), cultured them in vitro in IVC medium and analysed the frequency of cells displaying the mitotic marker, phosphorylated histone H3[19]. We found that the red fluorescent, diploid clone showed a higher mitotic index in the diploid (red)–aneuploid epiblast than in the diploid (red)–diploid epiblast

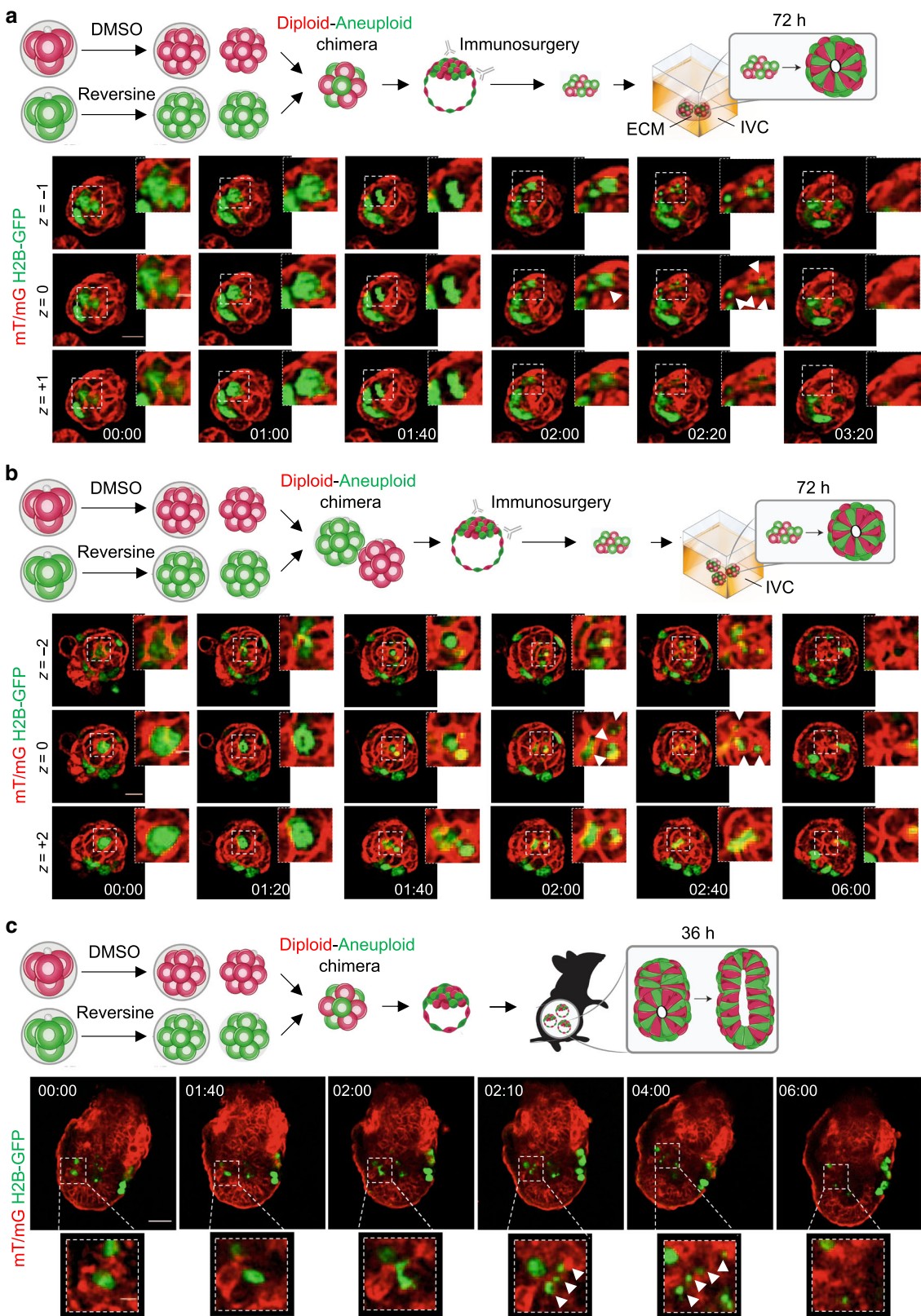

during the peri-implantation stage of development (Fig. 3d). These results indicate increased proliferation of the diploid cells in the presence of aneuploid cells, compensating for the loss of aneuploid cells in the diploid–aneuploid mosaic epiblast, thus allowing for the maintenance of a similar number of cells as found in diploid–diploid epiblasts.

**Autophagy is required for aneuploid cell elimination**. Since our results above together with our previous report[14], indicate that the aneuploid cells become eliminated during late pre- and peri-implantation developmental stages, we sought to investigate the mechanism triggering this process in more detail. We first wished to confirm the role of apoptosis in the disappearance of aneuploid

**Fig. 2 Elimination of aneuploid epiblast cells during peri-implantation development by apoptosis.** Chimeras containing a 1:1 ratio of control (diploid) and reversine-treated (aneuploid) cells were constructed from mT/mG (red) diploid cells and Histone H2B-GFP (green) aneuploid cells at the eight-cell stage and cultured beyond the blastocyst stage. Sequential representative images from time-lapse series for three diploid–aneuploid chimeras are shown, each showing apoptosis of an aneuploid cell (histone H2B-GFP) (white boxes) during pre- to post-implantation development. White arrows indicate the apoptotic debris. **a** Eight-cell diploid–aneuploid chimeras ($n = 12$ embryos) were generated at the eight-cell stage. Immunosurgery was performed at the late blastocyst stage to isolate the ICM from the TE. The ICMs were embedded in Matrigel and cultured in IVC medium for 72 h, during which they were live-imaged. Scale bar, 20 μm. Squares indicate magnified regions. Scale bar, 7 μm. Three z-planes have been shown. **b** Sixteen-cell diploid–aneuploid chimeras ($n = 22$ embryos) were generated at the eight-cell stage. Immunosurgery was performed at the late blastocyst stage and ICMs were cultured in IVC medium for 72 h, during which they were live-imaged. Scale bar, 20 μm. Squares indicate magnified regions. Scale bar, 7 μm. Three z-planes have been shown. **c** Eight-cell diploid–aneuploid chimeras ($n = 12$ embryos) were generated at the eight-cell stage. At the early blastocyst stage, the chimeras were transferred to pseudo-pregnant mothers and recovered 12 h after implantation to be cultured in vitro for 36 h and live-imaged. Scale bar, 40 μm. Squares indicate magnified regions. Scale bar, 10 μm.

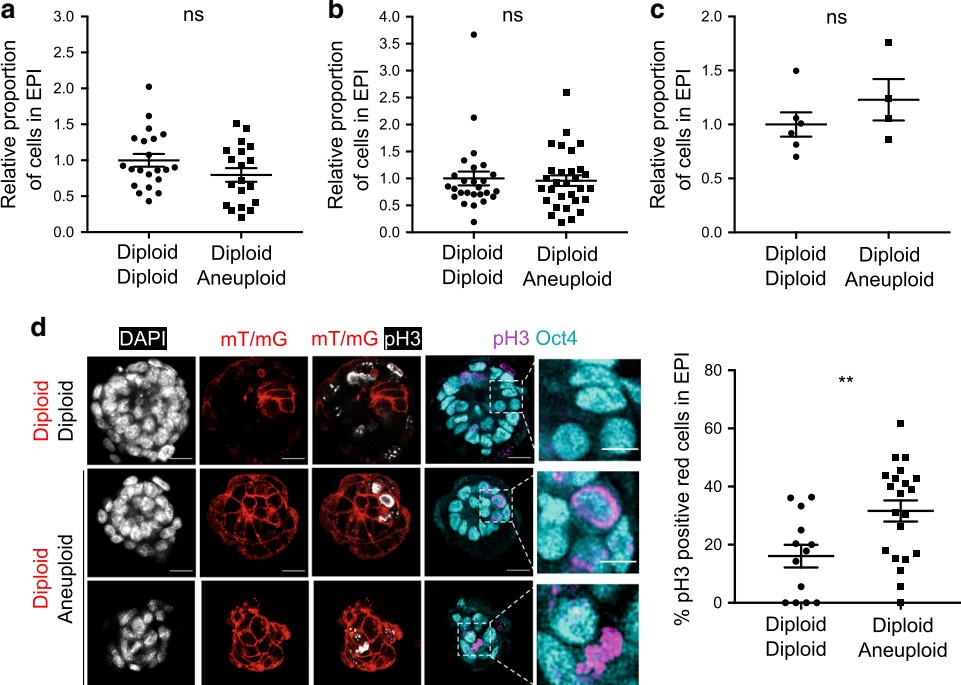

**Fig. 3 Size regulation of diploid–aneuploid epiblasts during peri-implantation development.** Embryos were treated with reversine (or DMSO) at the four- to eight-cell stage transition. **a** Eight-cell diploid–aneuploid and diploid–diploid chimeras were generated at the eight-cell stage. Immunosurgery was performed at the late blastocyst stage to isolate the ICM from the TE. The ICMs were embedded in Matrigel and cultured in IVC medium for 72 h. $n = 19$ diploid–aneuploid; $n = 1900$ EPI cells and $n = 21$ diploid–diploid chimeras; $n = 2636$ EPI cells. Student's t test. **b** Sixteen-cell diploid–aneuploid and diploid–diploid chimeras were generated at the eight-cell stage. Immunosurgery was performed on the double size chimeras and ICMs were cultured in IVC medium for 72 h as above. Diploid–diploid $n = 26$ chimeras; $n = 1772$ EPI cells and diploid–aneuploid $n = 30$ chimeras; $n = 1961$ EPI cells. Mann–Whitney test. **c** Eight-cell diploid–aneuploid and diploid–diploid chimeras were transferred to pseudo-pregnant mothers and recovered 12 h after implantation and then in vitro cultured for 36 h. $n = 4$ diploid–aneuploid; $n = 604$ EPI cells and $n = 6$ diploid–diploid chimeras; $n = 737$ EPI cells. Mann–Whitney test. For graphs **a–c**, relative number of cells in the EPI were analysed for both types of chimeras (relative to the average of diploid–diploid chimeras) at the end of the peri-implantation culture to investigate the level of size regulation of diploid–aneuploids with respect to diploid–diploids. ns = not significantly different. **d** Eight-cell diploid (red)–aneuploid and diploid (red)–diploid chimeras were generated. Immunosurgery was performed and ICMs were cultured as above for 48 h. The percentage of the number of pH3-positive red fluorescent EPI cells of the total red fluorescent EPI cells was analysed for each chimera, for both diploid–diploid and diploid–aneuploid ICMs. $n = 21$ diploid–aneuploid; $n = 546$ red EPI cells and $n = 13$ diploid–diploid chimeras; $n = 217$ red EPI cells. Scale bars, 20 μm. Squares indicate magnified regions. Scale bars, 10 μm. Student's t test and $**p = 0.0084$. For all the graphs, all data are mean ± s.e.m. Source data are provided as a Source Data file.

cells as the pre-implantation development progresses. To this end, we generated aneuploid embryos, by reversine treatment at the four- to eight-cell stage as before, and cultured aneuploid and diploid control embryos in the presence of the pan-caspase inhibitor ZVAD, which prevents apoptosis[20]. We imaged development of embryos in the presence of SYTOX[21], which allowed us to detect cell death in live embryos during blastocyst maturation (Supplementary Fig. 7a, Supplementary Movie 4). We found that ZVAD treatment reduced the number of dying cells in

the aneuploid embryos (Supplementary Fig. 7b). Consequently, the reduction in the epiblast cell number in aneuploid late blastocysts compared to diploid late blastocysts, was alleviated after ZVAD treatment (Supplementary Fig. 7c).

It has been shown that gene imbalances caused by aneuploidy can lead to proteomic imbalances[22–24]. We have therefore next set out to investigate whether aneuploid embryos display proteotoxic stress. As the heat shock protein 70 (HSP70) are a group of molecular chaperones instrumental in alleviating

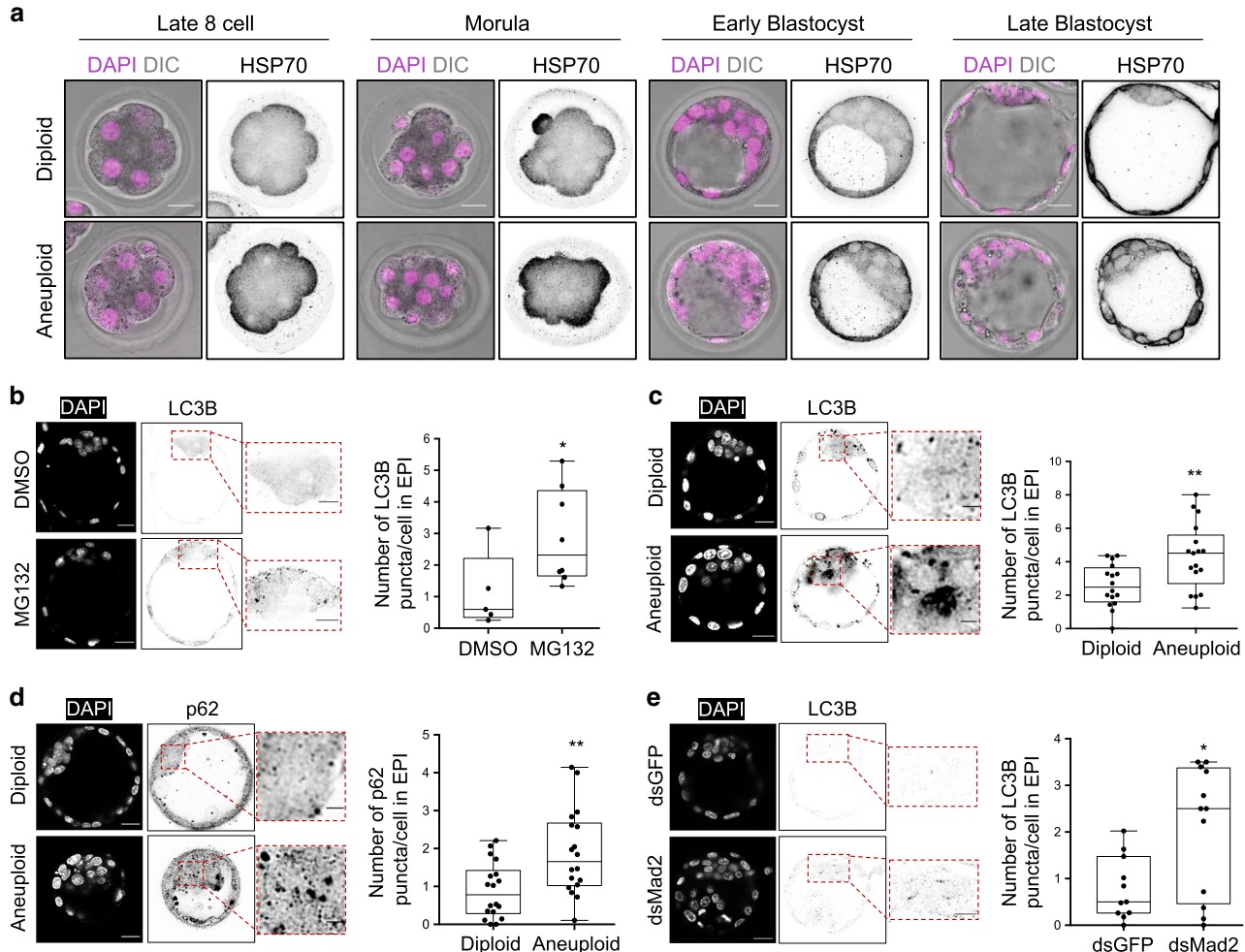

**Fig. 4 Chronic protein misfolding and autophagy upregulation in the aneuploid EPI cells. a** HSP70 immunostaining in diploid and aneuploid embryos at indicated stages. Scale bars, 20 μm. Control diploid embryos: 8-cell $n = 12$, morula $n = 9$, early blastocyst $n = 9$, late blastocyst $n = 14$. Reversine-treated aneuploid embryos: 8-cell $n = 10$, morula $n = 9$, early blastocyst $n = 9$, late blastocyst $n = 18$. **b** Embryos were treated with MG132 (or DMSO) at the late blastocyst stage for 6 h and immunostained for LC3B. Each dot represents the average number of LC3B puncta/cell in an embryo. Mann–Whitney test. Control $n = 5$, MG132-treated $n = 8$ embryos. Scale bars, 20 μm. Squares indicate the magnified regions. Scale bars, 10 μm. *$p = 0.0295$. Analysis of LC3B (**c**) and p62 (**d**) immunostaining in diploid and aneuploid late blastocysts' EPI. Each dot represents the average number of LC3B (**c**) or p62 (**d**) puncta/cell in an embryo. Scale bars, 20 μm. Squares indicate magnified regions. Scale bar, 5 μm. For **c**, diploid $n = 16$ embryos and aneuploid $n = 17$ embryos. Student's $t$ test, **$p = 0.0059$. For **d**, diploid $n = 18$ embryos and aneuploid $n = 18$ embryos. Student's $t$ test, **$p = 0.0033$. **e** Zygotes were injected with dsGFP (control) or dsMad2 and immunostained for LC3B at the late blastocyst stage. Each dot represents the average number of LC3B puncta/cell in an embryo. Student's $t$ test with Welch's correction, *$p = 0.0106$. dsGFP $n = 11$, dsMad2 $n = 12$ embryos. Scale bars, 20 μm. Squares indicate the magnified regions. Scale bars, 10 μm. For graphs **b–e**, data are shown as individual data points in a Box and Whiskers graph (bottom: 25%; top: 75%; line: median; whiskers: min to max). Source data are provided as a Source Data file.

misfolded protein stress[25], we compared HSP70 levels between diploid (control) and aneuploid (reversine-treated) embryos. We found that aneuploid embryos displayed higher HSP70 levels than diploid embryos from the eight-cell stage through to the late blastocyst stage (Fig. 4a), indicative of chronic misfolding. HSP70 levels in the epiblast of aneuploid late blastocysts were also significantly higher than in diploid blastocysts (Supplementary Fig. 8a).

The cell responds to an increase in misfolded proteins by upregulating protein quality control mechanisms such as autophagy[26]. Therefore, we hypothesised that autophagy might be upregulated in aneuploid embryos in response to this proteotoxic stress. During autophagy, the autophagosome forms around cargo and fuses with lysosomes, leading to degradation of the cargo[27]. The microtubule-associated light-chain 3B (LC3B) and p62/sequestosome 1 (SQSTM1) proteins are associated with the autophagosome

membrane and accumulate during autophagy[27]. They are the two widely used markers of autophagy[28]. Firstly, to investigate if proteotoxic stress can lead to the upregulation of autophagy, we treated embryos with 5 μM MG132, the proteasome inhibitor[29], for 6 h. We observed a significant increase in the LC3B accumulation in the MG132-treated epiblast compared to the control epiblast at the late blastocyst stage (Fig. 4b). To investigate the same in aneuploid cells, LC3B and p62 levels were analysed in the epiblast of aneuploid blastocysts. We observed a significant increase in the number of LC3B (Fig. 4c) and p62 (Fig. 4d) puncta in the epiblast of aneuploid blastocysts compared to diploid blastocysts. These increases in LC3B and p62 puncta in the aneuploid blastocysts compared to controls were at least in part due to increased levels of their respective transcripts (Supplementary Fig. 8b). As an alternative method to reversine, we also wished to inhibit the SAC by targeting the SAC protein Mad2[14], in order to determine whether the

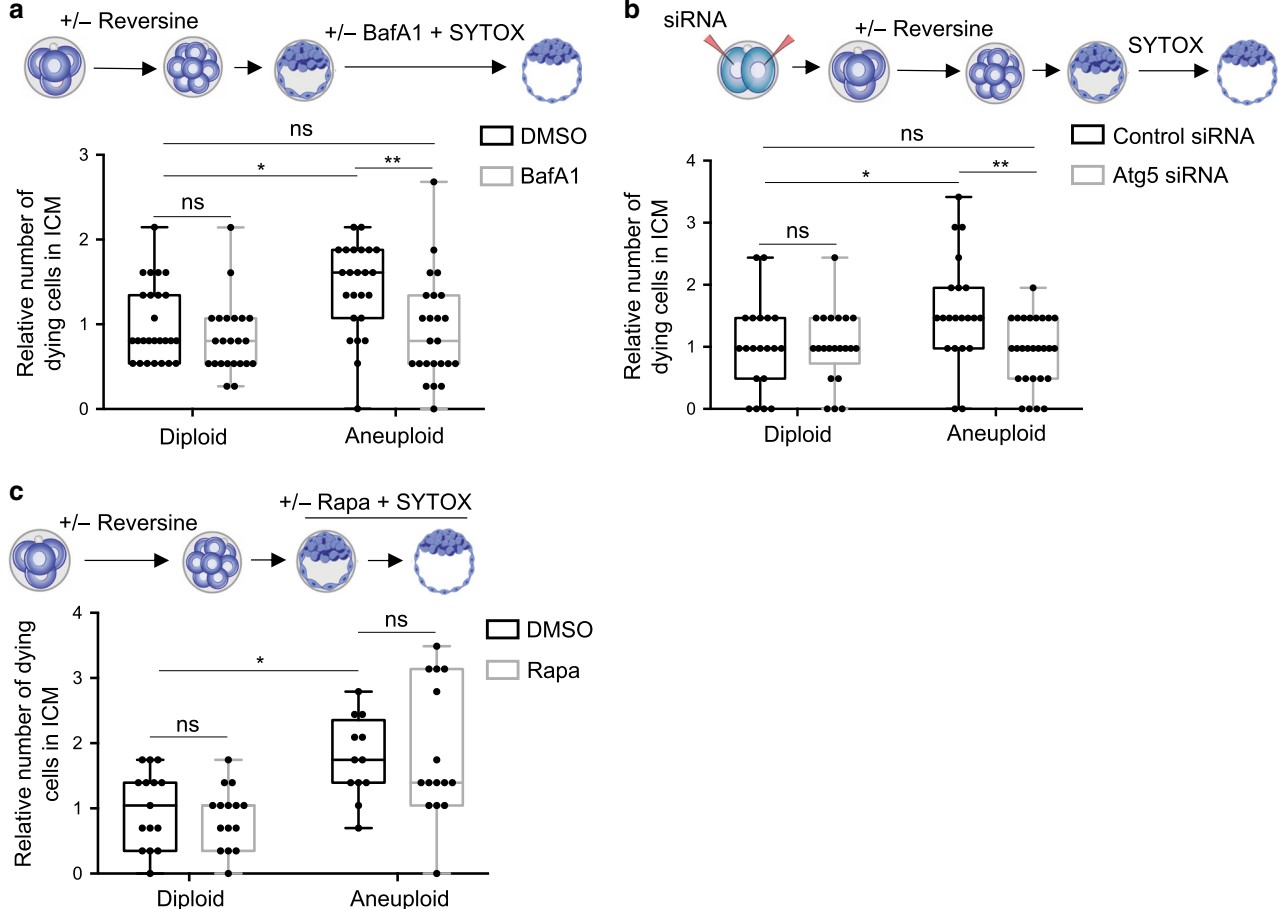

**Fig. 5 Autophagy upregulation mediates cell death in the ICM of aneuploid pre-implantation embryos. a** Diploid and aneuploid embryos were imaged with Bafilomycin A1 (BafA1) or DMSO and SYTOX from the early to late blastocyst stage (24 h). The number of dying ICM cells was assessed relative to the average number of dying cells in DMSO-treated diploid ICMs. Diploid $n = 26$ embryos, aneuploid $n = 24$ embryos, diploid BafA1 $n = 23$ embryos, aneuploid BafA1 $n = 24$ embryos. Kruskal–Wallis test, *$p = 0.0435$, **$p = 0.0050$. **b** Two-cell stage embryos were injected with Atg5 siRNA or control siRNA, treated at the four- to eight-cell stage with reversine or DMSO, and imaged with SYTOX during blastocyst maturation (24 h). The number of dying ICM cells was assessed relative to the average number of dying cells in control siRNA-injected diploid ICMs. Diploid $n = 20$ embryos, aneuploid $n = 21$ embryos, diploid Atg5 siRNA $n = 21$ embryos, aneuploid Atg5 siRNA $n = 27$ embryos. One-way ANOVA test, *$p = 0.0374$, **$p = 0.0068$. **c** Diploid (DMSO-treated) and aneuploid (reversine-treated) embryos were treated with DMSO or rapamycin (Rapa) from the early to the late blastocyst stage (24 h) and imaged in the presence of SYTOX to label dying cells. Diploid $n = 15$ embryos, aneuploid $n = 12$ embryos, diploid rapa $n = 15$ embryos, aneuploid rapa $n = 15$ embryos. One-way ANOVA test, *$p = 0.0358$. For all the graphs, data are shown as individual data points in a Box and Whiskers graph (bottom: 25%; top: 75%; line: median; whiskers: min to max), ns not significantly different. Source data are provided as a Source Data file.

upregulation of autophagy observed in reversine-treated epiblast cells could be an off-target effect of reversine. To this end, we injected zygotes with dsRNA targeting GFP (dsGFP), as a control, or Mad2 (dsMad2) and analysed LC3B accumulation at the blastocyst stage. We confirmed that injection of dsMad2 reduced *Mad2* mRNA to 15% relative to embryos injected with dsGFP (Supplementary Fig. 8c). We found that dsRNA-mediated depletion of Mad2 also led to significant increase in the LC3B accumulation in the epiblast (Fig. 4e). Overall, these results suggest that aneuploid epiblast cells upregulate autophagy at the blastocyst stage.

To investigate the possible role of autophagy in aneuploid embryos, we used the lysosomal inhibitor Bafilomycin A1[28] (BafA1) or RNAi-mediated depletion of the essential autophagy factor Atg5[30] to disrupt autophagy. We treated aneuploid and diploid embryos with 160.6 nM BafA1 and imaged them in the presence of SYTOX to detect dying cells from the early to the late blastocyst stage. We found that BafA1 treatment reduced the number of dying cells in the ICM of aneuploid, but not diploid, embryos (Fig. 5a). Similarly, we injected two-cell stage embryos with Atg5 siRNA, treated them with reversine or DMSO at the

four- to eight-cell stage and imaged them in the presence of SYTOX from the early blastocyst to the late blastocyst stage. We confirmed that injection of Atg5 siRNA reduced *Atg5* mRNA to 23% relative to embryos injected with control siRNA (Supplementary Fig. 8d). RNAi-mediated depletion of Atg5 also reduced the number of dying cells in the ICM of aneuploid, but not diploid, embryos (Fig. 5b). To further confirm the role of autophagy in the elimination of aneuploid cells, we treated embryos with rapamycin[31], which induces autophagy. We found that rapamycin treatment did not affect the number of dying cells in the ICM of either aneuploid or diploid embryos (Fig. 5c). Interestingly, rapamycin treatment did not increase the number of dying cells in the ICM of aneuploid embryos. This could be because the elimination of aneuploid cells from the mouse epiblast may not be dependent on the mTOR-autophagy pathway or alternatively, autophagy may be required but might not be sufficient to eliminate aneuploid cells. Future studies might be able to distinguish between these possibilities. Taken together our results suggest that autophagy is required to eliminate aneuploid ICM cells before implantation.

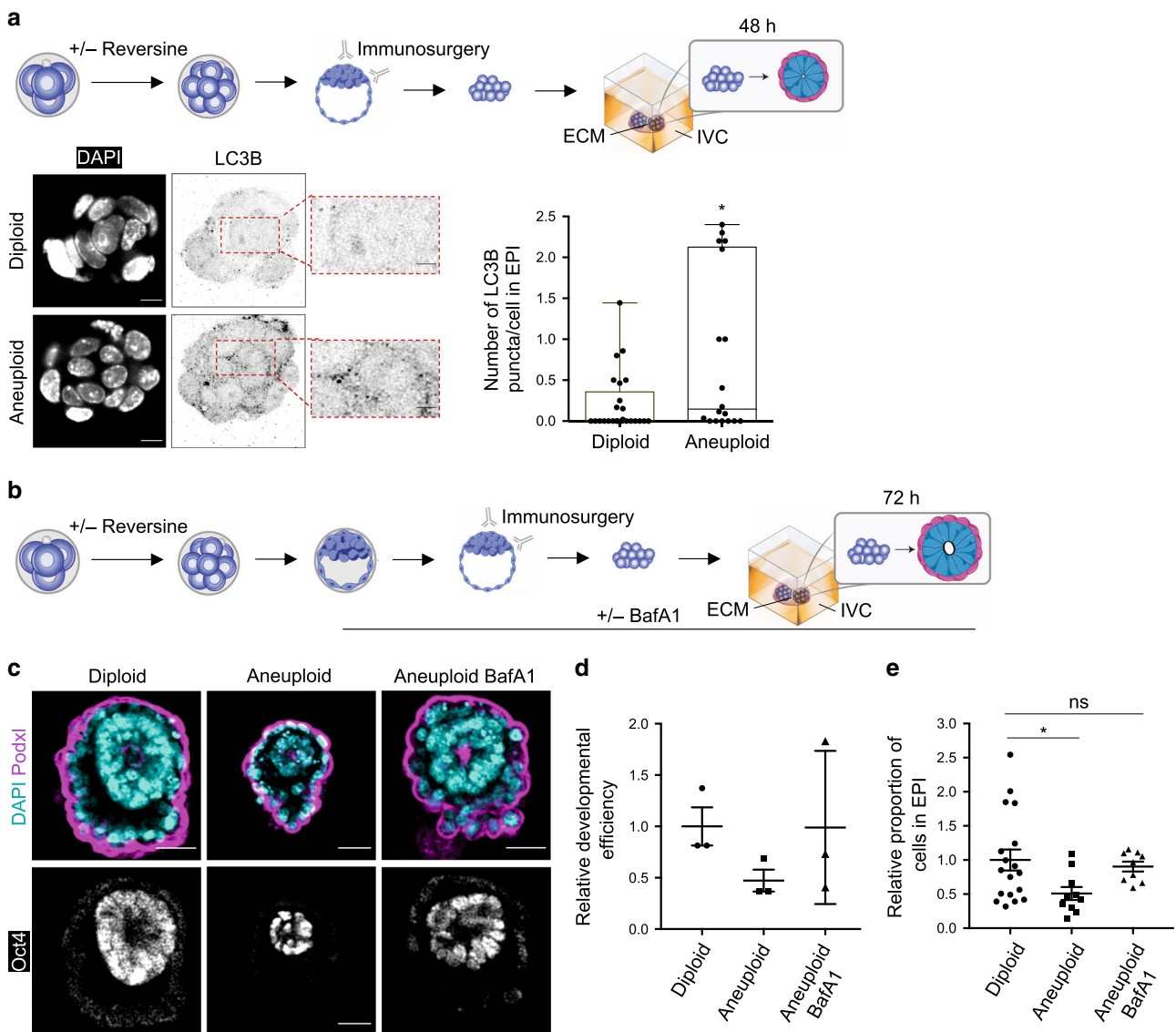

**Fig. 6 Autophagy upregulation mediates cell death in the ICM of aneuploid peri-implantation embryos. a** Embryos were treated at the four- to eight-cell stage with reversine or DMSO. Control (diploid) and reversine-treated (aneuploid). After immunosurgery at the late blastocyst stage, ICMs were embedded in Matrigel and cultured in IVC medium. LC3B immunostaining was analysed in diploid and aneuploid EPIs after 48 h IVC in vitro culture. Each dot represents the average number of LC3B puncta/cell in each EPI. Scale bars, 7 μm. Squares indicate magnified regions. Scale bars, 2 μm. Mann–Whitney test, *p = 0.0409. Diploid n = 25 embryos and aneuploid n = 18 embryos. Data are shown as individual data points in a Box and Whiskers graph (bottom: 25%; top: 75%; line: median; whiskers: min to max). **b** Embryos were treated at the four- to eight-cell stage with reversine or DMSO. Diploid and aneuploid embryos were cultured in DMSO or BafA1 during blastocyst maturation. Immunosurgery was performed and ICMs were cultured as above for 72 h in DMSO or BafA1. **c** Diploid and aneuploid ICMs were cultured as shown in (**b**) for 72 h and analysed for the efficiency of formation of an organised structure comprising an epithelised EPI surrounded by a PE layer with a central lumen. Scale bars, 30 μm. **d** Relative (to diploids) efficiency of ICMs in forming an organised structure, assessed according to (**c**), was evaluated. n = 3 independent experimental groups. Diploid n = 34 embryos, aneuploid n = 36 embryos and aneuploid BafA1 = 31 embryos. **e** Relative (to diploids) number of cells in the EPI was analysed for organised structures obtained in (**c**) for all three conditions. One-way ANOVA test, *p = 0.0450. For **c** and **e**, diploid n = 18 embryos; n = 1246 EPI cells, aneuploid n = 10 embryos; n = 352 EPI cells and Aneuploid BafA1 = 9 embryos; n = 563 EPI cells. For graphs **d** and **e**, all data are mean ± s.e.m. For all the graphs, ns not significantly different. Source data are provided as a Source Data file.

We next sought to investigate whether the elimination of aneuploid cells in peri-implantation embryos is triggered by the same molecular cascade as at the blastocyst stage. To this end, we treated embryos with reversine (or DMSO for control) at the four-to eight-cell stage transition, as before, performed immunosurgery to remove the TE, and cultured the ICMs for 48 h in IVC medium in Matrigel. We observed an increase in LC3B puncta in aneuploid embryos compared to diploid embryos, suggesting that autophagy is increased as the aneuploid ICM develops beyond the blastocyst stage (Fig. 6a).

To investigate the functional significance of autophagy in the elimination of aneuploid cells, we cultured aneuploid embryos in the presence of BafA1 from the early blastocyst to the late blastocyst stage, performed immunosurgery to isolate the ICMs, and cultured the ICMs in the presence of BafA1 for a further 72 h (Fig. 6b). We found that the inhibition of autophagy led to an

increase in the number of epiblast cells and allowed more embryos to form organised structures in which the primitive endoderm surrounded an epiblast which underwent lumenogenesis (Fig. 6c–e). Consequently, the size of the epiblast of diploid and autophagy-inhibited aneuploid rosette structures were equivalent (Fig. 6c, e), suggesting that autophagy is involved in elimination of aneuploid cells that otherwise would have contributed to development.

To confirm whether autophagy indeed led to the elimination of aneuploid cells, we again cultured reversine-treated embryos with (or without as controls) BafA1 from the early blastocyst stage to the early post-implantation stage and examined their chromosomal integrity by performing metaphase spreads. This revealed that while reversine-treated embryos without autophagy inhibition were mostly diploid, the reversine-treated embryos after autophagy inhibition were mostly aneuploid (Supplementary Fig. 8e). Taken together our results suggest that autophagy eliminates aneuploid cells at both the pre- and peri-implantation stages of development.

**p53 activation induces autophagy-mediated cell elimination**. Aneuploidy arises from chromosome missegregation. Studies of human cells and mouse embryos have shown that chromosome missegregation is followed by p53 activation, and that p53 limits the proliferation of aneuploid cells[32–34]. To test if the induction of aneuploidy induces a similar p53 response in the mouse embryo, we treated embryos with reversine (or DMSO) at the four- to eight-cell stage, as previously, and examined the mRNA levels of p53 and three p53-responsive genes[35], the cyclin-dependent kinase inhibitor *p21*, *cyclin G1* and *bcl-2*. We observed an increase in the mRNA levels of *p53*, *p21* and *cyclin G1*, and a decrease in *bcl-2* mRNA levels, in aneuploid blastocysts compared to diploid blastocysts, indicating an upregulation of the p53 pathway (Fig. 7a). As a positive control, embryos were treated with Nutlin-3[36], a p53-activating drug (or DMSO) from the late eight-cell stage until the late blastocyst stage and examined for *cyclin G1* mRNA levels. We observed an increase in *cyclin G1* mRNA levels in Nutlin-3 treated blastocysts compared to control blastocysts (Supplementary Fig. 9a).

To investigate the functional significance of p53 in the elimination of aneuploid cells, we used RNAi to deplete p53. p53 siRNA injection reduced *p53* mRNA to 16% and *cyclin G1* mRNA to 6.3% relative to embryos injected with control siRNA (Supplementary Fig. 9b). We injected both blastomeres at the two-cell stage with p53 siRNA, treated embryos with reversine or DMSO at the four- to eight-cell stage and imaged in the presence of SYTOX to detect dying cells. We found that p53 depletion reduced the number of dying cells in the ICM of aneuploid embryos (Fig. 7b), suggesting that the p53 pathway mediates the removal of aneuploid cells. Since p53 depletion also reduced the number of dying cells in the ICM of diploid embryos, this indicates that the elimination of cells by p53 pathway is not specific to the induction of chromosomal abnormality.

Since our results indicate that both the p53 pathway and autophagy are required to eliminate aneuploid cells from the ICM during blastocyst maturation (Figs. 5 and 7b), we finally sought to determine if the p53 pathway is required to induce autophagy in aneuploid cells. To this end, we injected both blastomeres at the two-cell stage with p53 siRNA, treated embryos with reversine (or DMSO) at the four- to eight-cell stage and analysed LC3B accumulation upon p53 depletion in aneuploid pre-implantation embryos. We found that depletion of p53 reduced the number of LC3B puncta/cell in the epiblast of aneuploid blastocysts, but not diploid blastocysts (Fig. 7c). In contrast, depletion of Atg5 by siRNA did not affect the mRNA levels of *p53* in aneuploid

blastocysts, indicating that aneuploidy-induced activation of p53 does not result from autophagy upregulation (Supplementary Fig. 10). Overall, these results indicate that the aneuploidy-induced accumulation of LC3B results from p53 activation. Together, our findings suggest that the p53 pathway induces autophagy-mediated elimination of aneuploid cells from the mouse epiblast and we propose that the chronic misfolded protein stress stimulates the p53 pathway in such cells (Fig. 7d).

## Discussion

In this study, we have made three principle discoveries: first, that aneuploid cells are preferentially eliminated by apoptosis during epiblast remodelling at the peri- and early post-implantation stages of development and therefore prior to gastrulation; second, that p53-induced autophagy is required to eliminate aneuploid cells and third, that normal diploid cells increase their proliferation rate to compensate for aneuploid cell elimination to regulate embryo size.

The preferential depletion of aneuploid cells from the epiblast via apoptosis throughout the peri-implantation and early post-implantation stages of development that we report here (Fig. 8a), suggests that the progressive clonal depletion of aneuploid cells might ensure the survival of healthy mosaic embryos[12,13]. This study directly demonstrates the progressive depletion of aneuploid cells by apoptosis during peri-implantation and early post-implantation embryogenesis. As distinct aneuploidies show different levels of proliferative disadvantage[37,38], this could account for the variation we see in the timing and extent of the depletion of aneuploid cells, thereby influencing the epiblast cell number and overall developmental success, as it has been shown that the successful post-implantation development correlates with the number of cells in the epiblast[39].

Despite the depletion of aneuploid cells as development progresses, we have found that diploid–aneuploid mosaic epiblasts have a similar size to diploid epiblasts by the early post-implantation stage. Since we find that diploid cells within mosaic epiblasts increase their proliferation rate, our results suggest that diploid cells compensate for the loss of aneuploid cells and in this way can regulate the size of the epiblast. This observation is in accord with reports showing that half-size embryos, made by separating blastomeres at the two-cell stage, can undergo compensatory proliferation to increase their size[40]. Although the mechanisms of size regulation remain unknown, this embryo plasticity is fundamental to tissue integrity and function.

To explore the relationship between aneuploidy and apoptosis, we compared the development of aneuploid and control embryos. Studies in diverse organisms indicate that overall gene expression and translation levels correlate with gene copy number in aneuploid cells[41–46]. Aneuploidy perturbs the normal stoichiometry of protein complex subunits, leading to intracellular proteomic imbalance[47]. Interestingly, aneuploidies of differing chromosomes have been reported to show common cellular responses including the downregulation of proliferation-related genes, the upregulation of energy metabolism and an increased sensitivity to conditions interfering with protein folding[41,48,49]. It has been shown in human cells that a persistent aneuploid karyotype, as in our case, is required to promote autophagy[50]. Here, we see a similar response and a significant upregulation of autophagy in the epiblast of aneuploid embryos.

To examine the involvement of autophagy in facilitating the elimination of aneuploid cells, we inhibited the process at two different stages: autophagosome formation by depleting Atg5 and lysosome fusion by treating with Bafilomycin A1. Both treatments attenuated cell death in the ICM of aneuploid embryos from the early to the late blastocyst stages suggesting that the entire

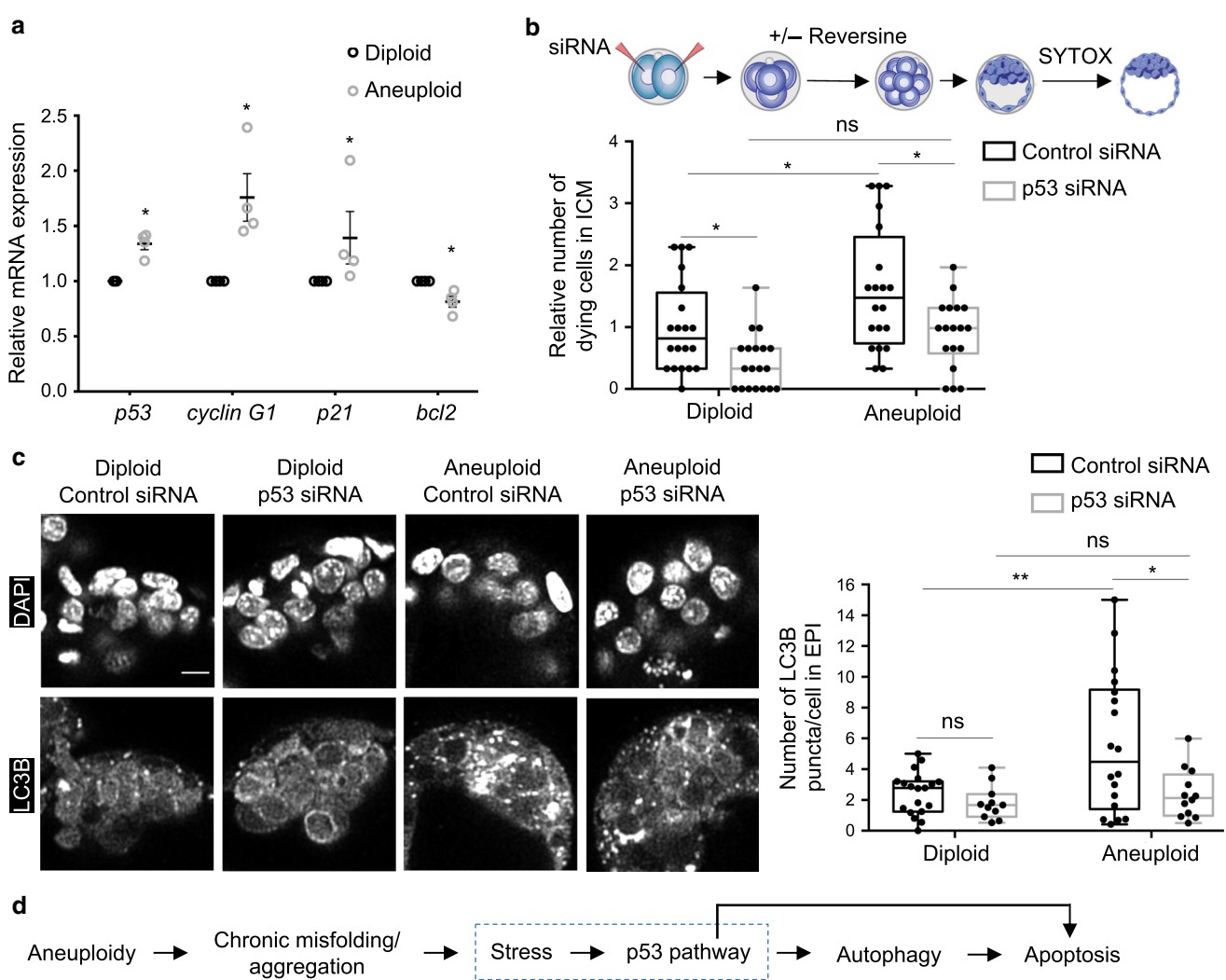

**Fig. 7 p53-induced autophagy in the ICM of aneuploid pre-implantation embryos. a** Embryos were treated at the four- to eight-cell stage with DMSO (diploid) or reversine (aneuploid) and mRNA expression for genes involved in p53 pathway were assessed at the late blastocyst stage (relative to diploid embryos) using qRT-PCR. Diploid $n = 69$ embryos, aneuploid $n = 67$ embryos. Mann–Whitney test, *$p = 0.0286$. All data are mean ± s.e.m. **b** Two-cell stage embryos were injected with p53 siRNA or control siRNA, treated from four- to eight-cell stage with reversine or DMSO, and imaged in the presence of SYTOX during blastocyst maturation (24 h). The number of dying cells in the ICM was assessed relative to the average number of dying cells in the ICM in control siRNA-injected diploids. Diploid $n = 20$ embryos, aneuploid $n = 20$ embryos, diploid p53 siRNA $n = 19$ embryos, aneuploid p53 siRNA $n = 18$ embryos. One-way ANOVA test, *$p = 0.0376$ (diploid versus aneuploid), 0.0476 (diploid vs. diploid p53 siRNA), 0.0158 (aneuploid vs. aneuploid p53 siRNA). **c** Two-cell stage embryos were injected with p53 siRNA or control siRNA, treated at the four- to eight-cell stage with reversine or DMSO. The level of autophagy (average LC3B puncta/cell in an embryo) in the EPI was assessed at the late blastocyst stage using immunostaining. Diploid $n = 19$ embryos, aneuploid $n = 18$ embryos, Diploid p53 siRNA $n = 11$ embryos, aneuploid p53 siRNA $n = 12$ embryos. Scale bar, 10 μm. One-way ANOVA test, **$p = 0.0055$, *$p = 0.0149$. For the graphs **b** and **c**, data are shown as individual data points in a Box and Whiskers graph (bottom: 25%; top: 75%; line: median; whiskers: min to max). For all the graphs, ns = not significantly different, *$p < 0.05$ and **$p < 0.01$. **d** Schematic of the events downstream of aneuploidy in mouse embryos, leading to programmed cell death of the cell. Source data are provided as a Source Data file.

autophagic flux is required. Inhibition of autophagy did not affect the level of cell death in untreated control embryos, consistent with the studies showing that autophagy is not involved in programmed cell death during embryo development[51,52]. While autophagy generally has pro-survival functions in a cell, it can occasionally lead to cell death[53]. The mechanisms that link autophagy and apoptosis are not clear but there are several potential routes through which autophagy could lead to apoptosis. Firstly, p62 could activate caspase-8 leading to further activation of apoptotic machinery. Secondly, autophagy might degrade cellular components leading the cell to eventually activate the apoptotic machinery. Finally, autophagy could specifically

degrade apoptotic machinery components (such as mitochondria) thereby lowering the pro-apoptotic activity threshold and activating apoptosis faster[53]. Since we observed that rapamycin treatment did not increase the number of dying cells in the aneuploid ICM, it will be interesting to explore in the future whether autophagy activates apoptotic machinery in aneuploid epiblast cells indirectly.

Our results also demonstrate that the p53-autophagy cascade contributes to the removal of aneuploid cells from the epiblast as we find that p53 depletion attenuates cell death in the ICM of aneuploid embryos during blastocyst maturation. We also show that autophagy upregulation is downstream of the p53 pathway in

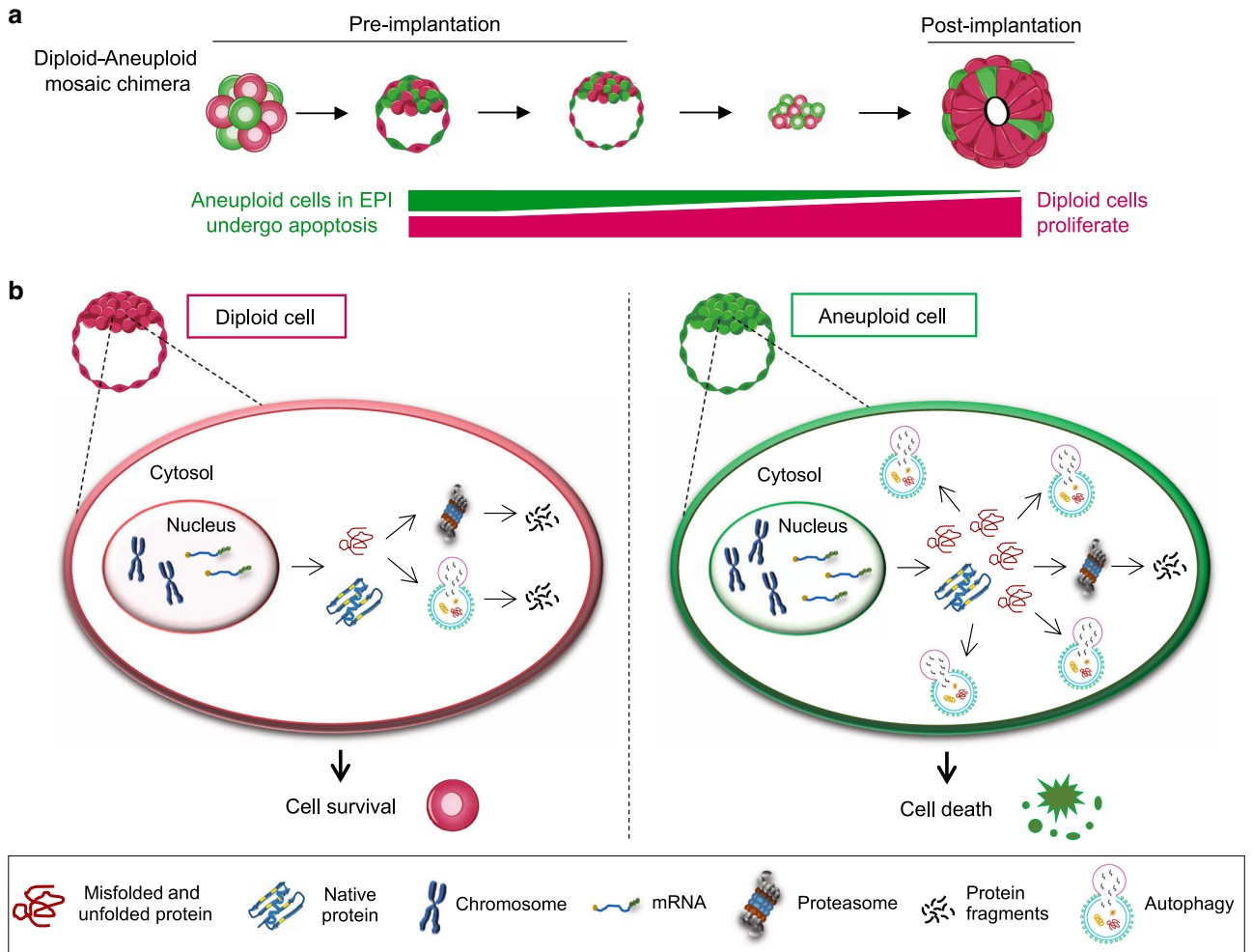

**Fig. 8 Model for the elimination of aneuploid cells in the mouse embryo. a** Aneuploid cells generated at the four- to eight-cell stage are progressively depleted from the epiblast of the mosaic embryo from the early blastocyst stage to the early post-implantation via apoptosis. Diploid cells in the same embryo over-proliferate to compensate for the reduction in overall epiblast cell number thereby allowing for successful development. **b** In a normal (diploid) cell, cellular protein quality control mechanisms, involving the proteasome machinery and autophagy, degrade misfolded/unfolded proteins to prevent cytotoxicity and promote healthy cell survival[26]. We hypothesise that in an aneuploid cell in the epiblast, gene aberrations are translated into protein aberrations. Chronic protein misfolding after several mitotic divisions upregulates autophagy to an extent where instead of protecting the cell, it mediates cell death. This prevents the aneuploid cell from continuing further in the development of the epiblast.

the epiblast of aneuploid embryos. However, the mechanisms which upregulate the p53 pathway in aneuploid embryos are not clear. In both yeast and mammalian cells, the bcl-2 anti-apoptotic protein can also function to prevent autophagy[54]. In accordance with this, we observe downregulation in the levels of bcl-2 in aneuploid embryos. In the future, it will be interesting to explore whether reduced levels of bcl-2 directly upregulate autophagy in aneuploid cells to mediate apoptosis.

This study establishes a direct link between autophagy and apoptosis in development of aneuploid embryos. Furthermore, this study provides direct evidence that autophagy in the epiblast can be utilised as a defence mechanism to deplete abnormal cells during late pre-, peri and early post-implantation development. While we expected aneuploid embryos to be eliminated at the early post-implantation stage, we saw a high incidence of aneuploidy in viable early post-implantation embryos when autophagy was inhibited. This indicates that if autophagy is defective or if it does not effectively deplete aneuploid cells, these cells would contribute to the subsequent embryo morphogenesis, with detrimental effect upon embryo survival to birth. In this study we

determined the effect of global aneuploidy on cell and embryo morphogenesis. Recently it was shown that tetraploid embryonic stem cells (ESCs) undergo apoptotic elimination from the mouse ESC–embryo chimeras via mTOR, when surrounded by diploid cells[20]. However, tetraploidy and single-chromosome aneuploidy could exhibit different physiological responses in a cell since tetraploidy quadruples the entire genome without affecting the relative stoichiometry of proteins, and therefore similar mechanisms might not apply to cell elimination in the model that we present here. A better understanding of the phenotypic effects of specific chromosomal abnormalities will help in future studies to further illuminate the cellular processes affected by aneuploidy. It will also be of future interest to explore whether similar mechanisms extend to human embryos and thereby further increase our understanding of early pregnancy loss.

## Methods

**Pre-implantation embryo culture and time-lapse imaging**. This research has been carried out following regulations of the Animals (Scientific Procedures) Act 1986—Amendment Regulations 2012—reviewed by the University of

Cambridge Animal Welfare and Ethical Review Body (AWERB). Animals were maintained in the Animal Facility at 12:12 light cycle and provided with food and water ad libitum. 4- to 6-week-old F1 (C57Bl/6× CBA) females were injected with 10 IU of pregnant mare's serum gonadotrophin (PMSG, Intervet) and, 48 h later, with 10 IU of human chorionic gonadotrophin (hCG, Intervet). These super-ovulated mice were then mated with F1 (C57Bl/6× CBA) males or, where indicated, with Histone H2B-GFP[55] or mT/mG (express membrane-targeted tandem dimer Tomato)[56] males. Embryos were recovered in M2 medium supplemented with 4 mg ml$^{-1}$ bovine serum albumin (BSA) and cultured in drops of KSOM media (Millipore) under mineral oil (Biocare Europe) in 37 °C and 5% CO$_2$. Reversine (Cayman Chemicals), Z-VAD-FMK (Enzo Life Sciences), Bafilomycin A1 (Sigma-Aldrich), Nutlin-3 (Cayman Chemicals), MG132 (Sigma-Aldrich) and Rapamycin (Millipore) were dissolved in dimethylsulfoxide (DMSO) (Sigma-Aldrich). They were respectively used at following final concentrations: 0.5 μM, 20 μM, 160.6 nM, 5 μM, 5 μM and 400 nM. Control embryos were incubated in the equivalent DMSO concentration. Analysis of cell death in the embryo was carried out using 5 μM SYTOX Orange nucleic acid stain (Life technologies). Embryos for live imaging were transferred to glass-bottom dishes (MatTek) and cultured within the individual interstices of a finely weaved nylon mesh (Plastok). Imaging was performed using a spinning disk confocal microscopy system (3i Intelligent Imaging Innovations) and SlideBook software. The images were captured every 10–20 min in 65 μm stacks of 2.0–2.5 μm intervals.

**Generation of chimeric embryos.** The zona pellucida of 8-cell stage embryos was removed by treatment with acidic Tyrode's solution (Sigma-Aldrich). Two types of mosaic chimeras were created at the eight-cell stage.

Single chimera: The embryos were incubated in Ca$^{2+}$/Mg$^{2+}$-free M2 for 5 min and then disaggregated into individual blastomeres by gentle mouth pipetting. Four control and four control/reversine-treated blastomeres were carefully aggregated together in M2 to get an eight-cell chimera.

Double size chimera: A control and a control/reversine-treated embryo were aggregated together in M2 to get a 16-cell chimera.

**Peri-implantation embryo culture and time-lapse imaging.** To culture embryos through the pre- to post-implantation transition, the zona pellucida was removed at the late blastocyst. Embryos/chimeras were exposed for 30 min to 20% rabbit anti-mouse whole serum (Sigma-Aldrich) in M2 medium at 37 °C. Next, they were incubated for 30 min with 20% guinea pig complement serum (Sigma-Aldrich) in M2 medium at 37 °C. The damaged TE was removed by pipetting the embryos in M2. ICMs were embedded in Matrigel. 20 μl drop of ice-cold growth factor-reduced Matrigel (BD Biosciences) was placed in a well of a μ-Slide 8-well ibiTreat (Ibidi) dish and embryos were mouth pipetted inside the Matrigel drop. The dish was incubated for 5 min at 37 °C to allow the Matrigel to solidify. Then, 300 μl of prewarmed IVC medium was added to the well.

Embryo imaging was performed using a spinning disk confocal at 37 °C and 5% CO$_2$. The images were captured every 20 min in 50–90 μm stacks of 2.5-μm intervals.

IVC medium constitution: Advanced DMEM F12 (Thermo Fisher Scientific), 20% v/v heat-inactivated fetal bovine serum (FBS) (Stem Cell Institute), GlutaMAX (Thermo Fisher Scientific), 25 U ml$^{-1}$ penicillin—25 μg ml$^{-1}$ streptomycin (Thermo Fisher Scientific), 1× ITS-X (Thermo Fisher Scientific), 8 nM β-oestradiol (Sigma-Aldrich), 200 ng ml$^{-1}$ progesterone (Sigma-Aldrich) and 25 μM N-aceyl-L-cysteine (Sigma-Aldrich).

**Post-implantation recovery and time-lapse imaging.** Chimeras at the blastocyst stage were transferred into the uterine horn of pseudo-pregnant females that had been mated with vasectomised males. Early post-implantation embryos were dissected from the maternal decidua and recovered into M2 medium. For live imaging, chimeras were transferred to glass-bottom dishes and cultured in drops of prewarmed IVC medium under mineral oil for 36 h. Imaging was performed using a spinning disk confocal microscopy system in 37 °C and 5% CO$_2$. The images were captured every 10 min in 70 μm stacks of 2.0-μm intervals.

**dsRNA preparation.** Mad2 dsRNA was designed to target a 300 bp region encompassing the Mad2 siRNA sequence described in Bolton et al. (2016) and was amplified from liver cDNA. In vitro transcription was performed using the MEGAscript T7 transcription kit (Thermo Fisher Scientific) following the manufacturer's instructions. The primer sequences used for dsMad2 preparation are as follows:

ATCGTAATACGACTCACTATAGGAAGCGTTTAAATGTAAACACAGCA
GAATCGTAATACGACTCACTATAGGCCTTTGTGGCTTGTGGTTGTAAATC

**Microinjection.** Microinjection was performed at the stage indicated using an Eppendorf Femtojet Microinjector[57]. Embryos were placed in drops of M2 medium covered by mineral oil during injection and transferred to pre-equilibrated KSOM

afterwards. Embryos were injected with 12 μM predesigned siRNAs (Qiagen) or with AllStars Negative Control (Qiagen) siRNA for control. For each gene, three oligos were supplied and mixed together in equal proportions. For dsRNA, embryos were injected with 700 ng μl$^{-1}$ dsRNA.

The siRNA sequences are as follows:
p53 siRNAs—CCGGGTGGAAGGAAATTTGTA, ACCGCCGTACAGAAGA AGAAA, TGGAGAGTATTTCACCCTCAA
atg5 siRNAs—ATGGTTCTAGATTCAATAATA, CAGAAGGTTATGAGACA AGAA, ACAGTTTGTATTTCTGATTAA

**qRT-PCR.** Blastocysts were collected for quantitative reverse transcriptase polymerase chain reaction (qRT-PCR) 96 h later. Total RNA was extracted using the Arcturus PicoPure RNA Isolation Kit and qRT-PCR was performed using the Power SYBR Green RNA-to-CT 1-Step Kit (Life Technologies) and a StepOne Plus Real-time PCR machine (Applied Biosystems). The following programme was used: 30 min 48 °C (reverse-transcription) followed by 10 min 95 °C followed by 45 cycles of 15 s 95 °C (denaturing) and 1 min 60 °C (annealing and extension). The ddCT method was used to determine relative levels of mRNA expression, with gapdh as an endogenous control. Primers are in Supplementary Table 1.

**Immunofluorescence.** Embryos were fixed in 4% PFA for 20 min at room temperature. The embryos were permeabilised, washed in PBST (0.1% Tween 20 (Sigma-Aldrich) in phosphate-buffered saline (PBS)) and incubated in blocking solution for 4 h at 4 °C and then with primary antibodies in blocking solution for overnight at 4 °C. They were then washed in PBST and incubated with Alexa Fluor secondary antibodies (Thermo Fisher Scientific, 1:400) in blocking solution for 1 h, washed again and incubated with DAPI (Thermo Fisher Scientific) for 5 min. Primary antibodies used: mouse anti-Cdx2 (Biogenex, 1:200), mouse anti-Oct3/4 (Santa Cruz Biotechnology, 5279, 1:200), goat Gata4 (Santa Cruz Biotechnology, 1237, 1:200), rat anti-Podocalyxin (R&D Systems, MAB1556, 1:500), rabbit anti-RFP (Rockland, 600-401-379, 1:500), mouse anti-HSP70 (Proteintech, 66183, 1:50), rabbit anti-p62 (Proteintech, 55274, 1:100), and rabbit anti-phospho-Histone H3 (Millipore, 06-570, 1:200). For primary antibody rabbit anti-LC3B (Cell Signalling, 2775, 1:200), embryos were fixed in 100% Methanol (Sigma-Aldrich) for 20 min at −20 °C. Confocal imaging was carried out using Leica SP5 (LAS AF software) inverted confocal microscope. Image files were viewed and analysed using ImageJ software.

For pre-implantation embryos: Permeabilisation was for 20 min in 0.5% Triton X-100 (Sigma-Aldrich) in PBS and blocking solution was 3% BSA (Sigma-Aldrich) in PBST.

For peri- or post-implantation embryos/ICMs: Permeabilisation was for 15 min in 0.3% Triton X-100 (Sigma-Aldrich) in PBS and blocking solution was 10% FBS in PBST.

**Metaphase spreads.** Embryos were cultured in 0.1 μg ml$^{-1}$ colcemid (Cayman chemical) for 12 h to arrest them in metaphase. Hypotonic solution: 1% sodium citrate (Sigma-Aldrich) and Carnoy's fixative: 3:1 methanol: glacial acetic acid (Sigma-Aldrich). Eight- to 16-cell embryos were incubated for 10 min in pre-warm hypotonic solution and fixed for 30 min in Carnoy's fixative. ICMs embedded in Matrigel were incubated in Cell Recovery Solution (Corning) for 20 min at 4 °C and then taken out of Matrigel using pipetting. They were incubated for 10 min in pre-warm hypotonic solution and fixed for 30 min in Carnoy's fixative.

Embryos were mouth pipetted on clean SuperFrost Plus Slide (Thermo Fisher Scientific). Slides were air-dried and mounted with ProLong® Gold Antifade Mountant with DAPI (Life Technologies). Spreads were analysed using Leica SP5 inverted confocal microscope.

**Statistics.** The statistical tests used are indicated in the corresponding figure legends. In all cases, the two-tailed version of the test was used. Normality of quantitative data was first assessed using D'Agostino's K-squared test. Statistical analysis was performed using Prism (GraphPad) software.

**Reporting summary.** Further information on research design is available in the Nature Research Reporting Summary linked to this article.

## Data availability

The authors claim that all relevant data of the findings in this work are provided within the paper and Supplementary Information files. Raw data are provided in the Source Data File.

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

## Acknowledgements

We are grateful to D. Glover, and M. Shahbazi for valuable comments on the paper; A. Weberling and C. Kyprianou for the post-implantation embryo recovery; A. Cox for the graphical representations; F. Antonica for help in the development of peri-implantation in vitro model. S.S. was supported by a Wellcome Trust PhD fellowship. This work was supported by Wellcome Trust (098287/Z/12/Z), ERC (669198), Rosetrees Trust (M877) and Open Philanthropy grants to M.Z.G.

## Author contributions

S.S. designed and conducted the experiments, analysed and interpreted the data with the help of L.K.I.-S. and M.Z. in some of the experiments. M.Z.-G. conceived and supervised the project and helped to interpret the data.

## Competing interests

The authors declare no competing interests.
