## [Peer Review File · Nature Communications]

Reviewers' Comments:

Reviewer #1:

Remarks to the Author:

This paper deals with a very important question in mammalian development: How are aneuploid cells eliminated during development? The authors previously showed that half of aneuploid cells in the mouse embryo remain following embryo implantation, so they now set out to establish what is the fate of these cells.

They performed an extensive series of experiments combining the use of chimeras, in vitro culture systems, and in vivo analyses of chimeric embryos transferred to uteri. The experiments involved complex embryological manipulation, as it had remained challenging to study these 'peri-implantation' stages of development (in previous studies, the authors developed many of the tools to tackle these technical challenges). They concluded that most aneuploid cells are eliminated during the time of epiblast remodeling via a combination of autophagy and apoptosis. Their results provide a timely characterization of aneuploidy dynamics in vivo (ie, when and where, and in which cell types it happens) and the mechanisms involved, and I support publication.

They examined the mechanistic basis underlying the elimination of this group of cells. They confirmed that the 'normal' cells compensate for the loss of the apoptotic cells, and they then go on to investigate the mechanism for this cell elimination. They applied drug treatment and molecular manipulations using siRNA approaches to show this is achieved via autophagy-like mechanisms. To further understand the mechanisms, they examined the involvement of p53. This revealed an increase in p53 levels during this process, and the authors then showed via p53 manipulations that the elimination of aneuploid cells is a p53-dependent event.

The precise molecular links between aneuploidy, apoptosis and autophagy remains somehow unclear, and it is likely to involve multiple cross-talking pathways. Yet their results provide valuable insights into the dynamics of aneuploid cell removal in mammalian development. Given the importance of chromosome segregation errors leading to aneuploidy for human health, the findings are of wide interest.

Overall the paper is straightforward, and the experiments are presented in a clear manner (their inclusion of multiple schemes also makes it easy to follow the experimental design for the various experiments).

It may be misleading however to refer to their reversine-treated and untreated cells as Aneuploid and Diploid cells. The authors state that: "...we will refer to reversine-treated cells as aneuploid and to control, DMSO-treated cells as diploid, throughout".

Control cells could still show aneuploidy (albeit at presumably lower frequency), so they cannot be considered 'necessarily' as diploid cells. The authors could consider using a more specific terminology, for example: "reversine-treated" and "reversine-untreated" cells.

Reviewer #2:

Remarks to the Author:

The paper by Singla and Zernicka-Goetz investigates whether and how aneuploid cells are eliminated from the embryonic lineage. By employing a mouse model of chromosome mosaicism, the authors make a compelling case for an active removal of cells with aneuploid karyotypes and demonstrate the existence of an autophagy-mediated process acting towards this goal. In particular, the authors make two important observations: 1. Aneuploid cells are actively removed in the embryo and, 2. Diploid cells are taking over in this process.

The paper provides an interesting perspective on the fate of aneuploid cells and the experiments are well executed. In general, the concepts presented are novel and the results are of interest for

the readership of Nature Communications. I have a few comments/suggestions.

Major points

The paper hinges on the exploitation of a well-established model of chromosome mis-segregation, namely reversine treatment. This reviewer would appreciate the usage of another method of induction of chromosome segregation errors (such as Mad2 RNAi) to validate at least the major findings.

The authors suggest that aneuploidy-induced proteotoxic stress activates autophagy, which ultimately leads to aneuploid cell removal. However, aneuploidy leads to several stresses and proteotoxicity is only one of them. Thus, it is important to validate the central role of proteotoxic stress by inducing it by other means and test whether it recapitulates the effects observed in aneuploid cells.

The first part of the results get bogged down in the details with a lengthy description of Supplementary Fig. 5. In contrast, the authors do not describe thoroughly the data presented in Figure 2, in which they present evidence for aneuploid cell removal by apoptosis. Since the data in figure 2 are crucial for the rest of the paper, this reviewer would like to see a more detailed description and discussion of the results presented in figure 2.

The authors state that "chronic misfolded protein stress stimulates the p53 pathway, which in turn induces autophagy-mediated elimination of aneuploid cells (Fig. 6d)". This statement is very strong and it is not supported by a mechanistic explanation. This reviewer thinks that either the authors provide molecular details for this or they tune down the conclusion.

Minor points

If autophagy is required to eliminate cells after implantation, Rapamycin treatment should decrease the number of aneuploid cells, since it induces autophagy. However, this does not seem the case in Supplementary Fig.8. Please explain.

It would be helpful to have a positive control for p53 activation for the data presented in Fig. 6A, such as Mdm2 inhibitors (e.g., Nutlin3) or DNA damage induction. This would give an idea of the dynamic range of p53 induction in the system.

Supp. Fig 1a: It is not stated how cells were arrested in metaphase

Supp. Fig 1c: it would be interesting to plot the data of chromosome counts as frequency plots, to have an idea of the different karyotypes obtained with reversine treatment

Reviewer #3:

Remarks to the Author:

1. In figure 2 authors are showing that aneuploidy cells are eliminated by the process of apoptosis. In the supplementary data authors are showing that the % of dying cells decreasing significantly in presence of apoptotic inhibitor in ICM but what about % of living cells in the ICM. As it will be increased whether those cells any defects on their growth pattern and cell shape? As apoptosis is a well-known process during development authors are requested to see the morphological changes of the living aneuploidy cells in the ICM.

2. In figure 3 authors are showing that LC3 puncta are accumulated in case of aneuploidy cells along with increase in p62 expression. It is inferred that autophagy is induced but there might be inhibition of fusion between autophagosome and lysosome as p62 expression is increased. How

will author justify the phenomenon? What is the status of autophagic flux during this process?

3. Authors are requested to observe the LC3 puncta accumulation in aneuploidy cells in bafilomycin-1 cells.

4. Authors are saying that inhibition of both apoptosis and autophagy leads to decrease in dying population of the aneuploidy cells. But as far as reports are concerned autophagy have dual role in regulation of cell death and also there is some crosslink between autophagy and apoptosis process. Here in this scenario the background of the study in relation to autophagy and apoptosis is too few. Authors are requested to explain them both by discussing the interdependence between the two phenomenon and establish the actual mechanism of cell death.

5. Authors are requested to see the proteosomal degradation of misfolded milieu.

Reviewer #4:

Remarks to the Author:

In their manuscript, Singla et al. explore the role of autophagy-induced apoptosis in the elimination of aneuploid cells during peri-implantation development.

To address this question, the authors' clever approach was based on previously published work and consists in using chimeras made of cells from normal embryos and embryos in which the proportion of aneuploid cells was increased using reversine, a potent inhibitor of the mitotic kinase Mps1. The authors then used a variety of approaches to test how aneuploid cells are eliminated: chimeras were either used in an in vitro ICM culture assay previously published by the authors or in embryo transfer experiments followed by in vitro cultures. The authors also used a wide array of techniques, including live imaging, small molecule inhibitors, siRNA microinjections, to probe how aneuploid cells are eliminated. The narrative of the manuscript, the topical issue and the varied technical approaches make for a compelling read.

However, at this stage, it is unclear whether the authors' findings overlap with previous work on cell competition in the pre- and early post-implantation mouse embryo (something that should be at least discussed).

In addition, I thought that, although most of the major conclusions made by the authors were supported by the data, there were specific places in the manuscript where the data was not entirely convincing and could be improved upon.

Please find below a list of points that I would like to put forward as I think they would strengthen the manuscript if addressed.

1. Results with proportion of labelled / non-labelled cells

There is a certain proportion of diploid-aneuploid chimeras that end up with 100% of non-fluorescent cells coming from reversine-treated embryos (Figure S3 and S4). In some cases, the frequency is similar to diploid-diploid chimeras (Figure S4D). This seems counter to the idea that aneuploid cells are eliminated from the embryos. Is this due to technical limitations related to the reversine treatment? Maybe some embryos do not see their proportion of aneuploid cells increase after reversine treatment?

Also, unless I am mistaken, there is does not seem to be any statistical analysis for the results presented in Figure S3E and F and S4D and E. It would be important to see if the distribution of frequencies is different between diploid-diploid and diploid-aneuploid chimeras.

2. Embryo transfer experiments

The authors use embryo transfers as a more "in vivo" approach to complement ICM in vitro cultures. However, the embryos were only transferred for 12 hours and this was followed by a further 36 hours in culture. This makes it hard to talk about "in vivo" conditions and I think it would therefore be appropriate to slightly reword some of the conclusions such as "Taken

together, these results indicate that aneuploid cells are preferentially eliminated during remodeling of the epiblast, both in vitro and in vivo". Also, I do wonder why the authors decided to culture the embryos after transfer considering they could have let them develop longer in vivo in some cases. Since they had to grow the embryos in vitro for live imaging experiments, I guess it might have been to keep the same conditions across several experiments. However, it would be good if this could be clarified.

3. Live imaging experiments

These experiments are understandably difficult to undertake. However, I thought that the results presented by the authors in figure 2 were currently of relatively limited added value for the following reasons:

- From the data provided, it is not always clear that the cells shown are dying of apoptosis. For example, the cell shown in Figure 2A looks like it could be dividing and then move out of the plane.
- There is no n number for Figure 2C and the embryo shown looks unhealthy, suggesting that the culture or imaging conditions may not be optimal.
- There is no quantification. There also appears to be a significant number of green cells that do not commit apoptosis during the duration of the experiment

4. Relative proportion of cells in epiblast (Figure 3A-C)

As I understand, the absolute number of cells was counted in the epiblast of diploid-diploid and diploid-aneuploid chimeras and this number was divided by the average number of cells in the epiblast of diploid-diploid chimeras. I think it would be good to clarify the rationale behind this decision.

5. The pH3 immunofluorescence in Figure 3D does not look like it has worked very well and it would be reassuring if the authors could show better examples of pH3-positive cells.

6. The immunofluorescence data for HSP70, LC3B and in particular p62 are unclear. HSP70 signal is strangely localised subapically in blastomeres, is that to be expected? The p62 aggregates are worryingly also present in the blastocyst cavity. Some additional controls (using siRNA or blocking peptides for example) would help clarify what is signal from noise.

Reviewer #1 (Remarks to the Author):

This paper deals with a very important question in mammalian development: How are aneuploid cells eliminated during development? The authors previously showed that half of aneuploid cells in the mouse embryo remain following embryo implantation, so they now set out to establish what is the fate of these cells.

They performed an extensive series of experiments combining the use of chimeras, in vitro culture systems, and in vivo analyses of chimeric embryos transferred to uteri. The experiments involved complex embryological manipulation, as it had remained challenging to study these 'peri-implantation' stages of development (in previous studies, the authors developed many of the tools to tackle these technical challenges). They concluded that most aneuploid cells are eliminated during the time of epiblast remodeling via a combination of autophagy and apoptosis. Their results provide a timely characterization of aneuploidy dynamics in vivo (ie, when and where, and in which cell types it happens) and the mechanisms involved, and I support publication.

They examined the mechanistic basis underlying the elimination of this group of cells. They confirmed that the 'normal' cells compensate for the loss of the apoptotic cells, and they then go on to investigate the mechanism for this cell elimination. They applied drug treatment and molecular manipulations using siRNA approaches to show this is achieved via autophagy-like mechanisms. To further understand the mechanisms, they examined the involvement of p53. This revealed an increase in p53 levels during this process, and the authors then showed via p53 manipulations that the elimination of aneuploid cells is a p53-dependent event.

The precise molecular links between aneuploidy, apoptosis and autophagy remains somehow unclear, and it is likely to involve multiple cross-talking pathways. Yet their results provide valuable insights into the dynamics of aneuploid cell removal in mammalian development. Given the importance of chromosome segregation errors leading to aneuploidy for human health, the findings are of wide interest.

Overall the paper is straightforward, and the experiments are presented in a clear manner (their inclusion of multiple schemes also makes it easy to follow the experimental design for the various experiments).

Response: We are very grateful to the referee for carefully reading our manuscript and for these very positive comments.

It may be misleading however to refer to their reversine-treated and untreated cells as Aneuploid and Diploid cells. The authors state that: "...we will refer to reversine-treated cells as aneuploid and to control, DMSO-treated cells as diploid, throughout".

Control cells could still show aneuploidy (albeit at presumably lower frequency), so they cannot be considered 'necessarily' as diploid cells. The authors could consider using a more specific terminology, for example: "reversine-treated" and "reversine-untreated" cells.

Response: We considered new terminology, but this was hindering the presentation of the data in a clear way and thus we reverted to the previous nomenclature for simplicity. However, with this comment in mind, we are now stating in the manuscript '... we induced chromosome segregation errors by treating embryos with a well-established small reversible inhibitor, reversine, during 4–8 cell division to inactivate the SAC¹⁵ and confirmed that this treatment significantly increased the incidence of aneuploidy in comparison to DMSO-treated controls (Supplementary Fig. 1a–d), in agreement with previous results¹⁴. Thus, we will refer to reversine-treated cells as aneuploid and to control, DMSO-treated cells as diploid throughout for simplicity.' (page 2).

Reviewer #2 (Remarks to the Author):

The paper by Singla and Zernicka-Goetz investigates whether and how aneuploid cells are eliminated from the embryonic lineage. By employing a mouse model of chromosome mosaicism, the authors make a compelling case for an active removal of cells with aneuploid karyotypes and demonstrate the existence of an autophagy-mediated process acting towards this goal. In particular, the authors make two important observations: 1. Aneuploid cells are actively removed in the embryo and, 2. Diploid cells are taking over in this process.

The paper provides an interesting perspective on the fate of aneuploid cells and the experiments are well executed. In general, the concepts presented are novel and the results are of interest for the readership of Nature Communications. I have a few comments/suggestions.

Response: We are very grateful to the referee for carefully reading our manuscript and for these supportive comments.

Major points

The paper hinges on the exploitation of a well-established model of chromosome mis-segregation, namely reversine treatment. This reviewer would appreciate the usage of another method of induction of chromosome segregation errors (such as Mad2 RNAi) to validate at least the major findings.

Response: We agree with the referee and indeed when we first used reversine as an inducer of chromosome aneuploidy, we also carried out experiments with Mad2 siRNA and obtained the same results. We found that Mad2 siRNA-injected embryos show a similar pre-implantation phenotype to that seen with reversine treatment; however, the effect was weaker¹⁴ and this is why we have carried out this study using reversine. Since Mad2 siRNA and reversine had similar pre-implantation phenotypes (a significant decrease in the epiblast cell number), we used reversine treatment as the method to induce aneuploidy in all the experiments for this paper. However, to validate the major finding in this paper from reversine-treated embryos, we have carried out new experiments in which we injected embryos with dsRNA targeting GFP (dsGFP), as a control, or Mad2 (dsMad2) at the zygote stage to inhibit the spindle assembly checkpoint (SAC) and induce aneuploidy through an alternative method. We then analysed embryos for LC3B (a marker of autophagy) accumulation in the epiblast, using immunofluorescence. We observed a significant increase in LC3B protein accumulation in the dsMad2-injected epiblast compared to dsGFP-injected epiblasts (Supplementary Fig. 8e). These results indicate that aneuploid mouse epiblast cells upregulate autophagy. The efficiency of dsMad2 was also assessed by qRT-PCR. We injected dsGFP (control) or dsMad2 at the 2-cell stage and then performed qRT-PCR at the 8-cell stage. We observed an 85% decrease in *Mad2* mRNA levels in the dsMad2-injected embryos compared to the dsGFP-injected embryos (Supplementary Fig. 8d). (page 8)

Supp. Figure 8d and e. (d) Both blastomeres at the 2-cell stage were injected with dsRNA targeting GFP (dsGFP), as a control or Mad2 (dsMad2). *Mad2* mRNA expression was assessed by qRT-PCR at the 8-cell stage (relative to dsGFP-injected embryos). dsGFP n = 15, dsMad2 n = 15 embryos. All data are

mean. (e) Zygotes were injected with dsGFP (control) or dsMad2 and immunostained for LC3B at the late blastocyst stage. Each dot represents the average number of LC3B puncta/cell in an embryo. Student's t-test with Welch's correction. dsGFP n = 11, dsMad2 n = 12 embryos. Scale bars, 20µm. Squares indicate the magnified regions. Scale bars, 10µm. Data are shown as individual data points in a Box and Whiskers graph (bottom: 25%; top: 75%; line: median; whiskers: min to max).

The authors suggest that aneuploidy-induced proteotoxic stress activates autophagy, which ultimately leads to aneuploid cell removal. However, aneuploidy leads to several stresses and proteotoxicity is only one of them. Thus, it is important to validate the central role of proteotoxic stress by inducing it by other means and test whether it recapitulates the effects observed in aneuploid cells.

Response: We agree with the referee and in response to the referee's comment, we have carried out new experiments in which we treated embryos for 6 hours at the late blastocyst stage with the proteasome inhibitor MG132 (5 µM) to elicit proteotoxic stress or DMSO (control) and then analysed embryos for LC3B (a marker of autophagy) accumulation in the epiblast, using immunofluorescence. We observed a significant increase in LC3B protein accumulation in the MG132-treated epiblast compared to control epiblasts (Supplementary Fig. 8b). These results suggest that proteotoxic stress can lead to upregulation of autophagy, supporting our conclusion that increased proteotoxic stress leads to an upregulation of autophagy in the epiblast of aneuploid mouse embryos. (page 7)

The first part of the results get bogged down in the details with a lengthy description of Supplementary Fig. 5. In contrast, the authors do not describe thoroughly the data presented in Figure 2, in which they present evidence for aneuploid cell removal by apoptosis. Since the data in figure 2 are crucial for the rest of the paper, this reviewer would like to see a more detailed description and discussion of the results presented in figure 2.

Response: We agree with the referee and now provide a more detailed description and discussion of the results presented in the Figure 2 (page 5). We have also added two additional 'z' planes for panels a and b in the Figure 2 to make the apoptosis and removal of the aneuploid cell more evident. We say: 'These morphological features indicative of apoptosis included nuclear condensation, followed by the formation of the apoptotic bodies and subsequent removal of the cellular debris (Fig. 2a, Supplementary Movie 1). We found that double size diploid-aneuploid chimeras also displayed similar features indicative of apoptosis of the aneuploid cells in the epiblast and engulfment of the apoptotic bodies by the neighbouring red fluorescent control cells (Fig. 2b, Supplementary Movie 2).' We therefore conclude this section by saying: 'These observations suggest that aneuploid cells are preferentially depleted from the epiblast by apoptosis and the apoptotic debris was gradually cleared from the embryo during the peri-implantation stages of development both *in vivo* and *in vitro*'.

The authors state that "chronic misfolded protein stress stimulates the p53 pathway, which in turn induces autophagy-mediated elimination of aneuploid cells (Fig. 6d)". This statement is very strong and it is not supported by a mechanistic explanation. This reviewer thinks that either the authors provide molecular details for this or they tune down the conclusion.

Response: As suggested, we tuned down our statement in the revised manuscript to read: 'our findings suggest that the p53 pathway induces autophagy-mediated elimination of aneuploid cells from the mouse epiblast and we propose that the chronic misfolded protein stress stimulates the p53 pathway in such cells (Fig. 6d).' (page 10).

Minor points

If autophagy is required to eliminate cells after implantation, Rapamycin treatment should decrease the number of aneuploid cells, since it induces autophagy. However, this does not seem the case in Supplementary Fig.8. Please explain.

Response: Thank you for this comment. We now discuss two of the possible explanations for this result. Firstly, as rapamycin induces autophagy via mTOR activation, the elimination of aneuploid cells from the mouse epiblast may not be dependent on the mTOR– autophagy pathway. Secondly, it may be that autophagy is required but is not sufficient to eliminate aneuploid cells and so increasing autophagy via rapamycin treatment is not sufficient to increase the elimination of aneuploid cells. We now explain this result in our manuscript by saying: ‘Interestingly, rapamycin treatment did not increase the number of dying cells in the ICM of aneuploid embryos. This could be because the elimination of aneuploid cells from the mouse epiblast may not be dependent on the mTOR– autophagy pathway or alternatively, autophagy may be required but might not be sufficient to eliminate aneuploid cells’. (page 8)

It would be helpful to have a positive control for p53 activation for the data presented in Fig. 6A, such as Mdm2 inhibitors (e.g., Nutlin3) or DNA damage induction. This would give an idea of the dynamic range of p53 induction in the system.

Response. To respond to the referee’s suggestion, we have carried out experiments in which we treated embryos from the late 8–cell stage to the late blastocyst stage with the Mdm2 inhibitor Nutlin–3 (5 μ M) or DMSO (as control) and then analysed them using qRT–PCR. We observed an increase in *cyclin G1* mRNA levels in Nutlin–3 treated blastocysts compared to the control blastocysts (Supplementary Fig. 9a). The increase in *cyclin G1* mRNA levels after Nutlin–3 treatment is comparable to the increase in *cyclin G1* mRNA levels in aneuploid embryos compared to diploid embryos. We state these results and say: ‘As a positive control, embryos were treated with Nutlin–3³⁶, a p53–activating drug (or DMSO) from the late 8–cell stage until the late blastocyst stage and examined for *cyclin G1* mRNA levels. We observed an increase in *cyclin G1* mRNA levels in Nutlin–3 treated blastocysts compared to control blastocysts (Supplementary Fig. 9a).’ (page 9)

Supp. Figure 9a. Embryos were treated with Nutlin–3 (or DMSO) from the late 8–cell to the late blastocyst stage. mRNA expression for the gene *cyclin G1* was assessed by qRT–PCR at the late blastocyst stage (relative to control embryos). All data are mean \pm s.e.m. Control n=29, Nutlin–3–treated n=30 embryos.

Supp. Fig 1a: It is not stated how cells were arrested in metaphase

Response: Cells were arrested in metaphase using 12 h colcemid treatment (we mentioned this information in the Material and Method section).

Supp. Fig 1c: it would be interesting to plot the data of chromosome counts as frequency plots, to have an idea of the different karyotypes obtained with reversine treatment

Response: As suggested, we now present this data in Supplementary Figure 1d (page 2).

Supp. Figure 1d. Distribution of reversine-treated cells that were aneuploid according to the number of chromosomes (n).

Reviewer #3 (Remarks to the Author):

1. In figure 2 authors are showing that aneuploidy cells are eliminated by the process of apoptosis. In the supplementary data authors are showing that the % of dying cells decreasing significantly in presence of apoptotic inhibitor in ICM but what about % of living cells in the ICM. As it will be increased whether those cells any defects on their growth pattern and cell shape? As apoptosis is a well-known process during development authors are requested to see the morphological changes of the living aneuploidy cells in the ICM.

Response: Our results show that the aneuploid cells which are not eliminated by the end of pre-implantation, continue to be depleted during implantation or early post-implantation development by apoptosis. In this current study we have focused specifically on the mechanism of elimination of aneuploid cells rather than the effect of retaining an increased proportion of aneuploid cells on embryonic development, which was a specific focus of our previous study (Bolton *et al.* Nature Communication, 2016).

2. In figure 3 authors are showing that LC3 puncta are accumulated in case of aneuploidy cells along with increase in p62 expression. It is inferred that autophagy is induced but there might be inhibition of fusion between autophagosome and lysosome as p62 expression is increased. How will author justify the phenomenon? What is the status of autophagic flux during this process?

[Redacted]

3. Authors are requested to observe the LC3 puncta accumulation in aneuploidy cells in bafilomycin-1 cells.

Response: Addressed in the previous comment.

4. Authors are saying that inhibition of both apoptosis and autophagy leads to decrease in dying population of the aneuploidy cells. But as far as reports are concerned autophagy have dual role in regulation of cell death and also there is some crosslink between autophagy and apoptosis process. Here in this scenario the background of the study in relation to autophagy and apoptosis is too few. Authors are requested to explain them both by discussing the interdependence between the two phenomenon and establish the actual mechanism of cell death.

Response: We thank the referee for this comment, which we addressed by expanding the discussion in our revised manuscript to say ‘While autophagy generally has pro-survival functions in a cell, it can occasionally lead to cell death⁵³. The mechanisms that link autophagy and apoptosis are not clear but there are several potential routes through which autophagy could lead to apoptosis. Firstly, p62 could activate caspase-8 leading to further activation of apoptotic machinery. Secondly, autophagy might degrade cellular components leading the cell to eventually activate the apoptotic machinery. Finally, autophagy could specifically degrade apoptotic machinery components (such as mitochondria) thereby lowering the pro-apoptotic activity threshold and activating apoptosis faster⁵³. Since we observed that rapamycin treatment did not increase the number of dying cells in the aneuploid ICM, it will be interesting to explore in the future whether autophagy activates apoptotic machinery in aneuploid epiblast cells indirectly.’ (Page 12).

5. Authors are requested to see the proteosomal degradation of misfolded milieu.

Response: Several studies using cellular models, mentioned in two papers below, have shown that proteasomal degradation machinery is compromised in aneuploid cells and is unable to degrade misfolded proteins:

1. Oromendia, A. B., Dodgson, S. E. and Amon, A. Aneuploidy causes proteotoxic stress in yeast. *Genes Dev.* **26**, 2696–2708 (2012).
2. Aivazidis, S. *et al.* The burden of trisomy 21 disrupts the proteostasis network in Down syndrome. *PLoS ONE* **12**(4), e0176307 (2017).

Our results indicate that the aneuploid epiblast cells in the mouse embryos exhibit similar effect (as suggested by the chronic increase in misfolded proteins) and upregulate autophagy in response to proteotoxic stress. We have performed additional experiments where we compromised proteasomal machinery by other means and tested whether it recapitulates the effects observed in aneuploid cells. We treated embryos for 6 hours at the late blastocyst stage with the proteasome inhibitor MG132 (5 μ M) to elicit proteotoxic stress or DMSO (control). We saw that the inactivation of proteasomal degradation in mouse blastocysts leads to the upregulation of autophagy in the epiblast, as shown now in Supplementary Figure 8b. (page 7)

Supp. Figure 8b. Embryos were treated with MG132 (or DMSO) at the late blastocyst stage for 6 h and immunostained for LC3B. Each dot represents the average number of LC3B puncta/cell in an embryo. Mann–Whitney test. Control n=5, MG132-treated n=8 embryos. Scale bars, 20µm. Squares indicate the magnified regions. Scale bars, 10µm. Data are shown as individual data points in a Box and Whiskers graph (bottom: 25%; top: 75%; line: median; whiskers: min to max).

Reviewer #4 (Remarks to the Author):

In their manuscript, Singla et al. explore the role of autophagy-induced apoptosis in the elimination of aneuploid cells during peri-implantation development.

To address this question, the authors' clever approach was based on previously published work and consists in using chimeras made of cells from normal embryos and embryos in which the proportion of aneuploid cells was increased using reversine, a potent inhibitor of the mitotic kinase Mps1. The authors then used a variety of approaches to test how aneuploid cells are eliminated: chimeras were either used in an in vitro ICM culture assay previously published by the authors or in embryo transfer experiments followed by in vitro cultures. The authors also used a wide array of techniques, including live imaging, small molecule inhibitors, siRNA microinjections, to probe how aneuploid cells are eliminated. The narrative of the manuscript, the topical issue and the varied technical approaches make for a compelling read.

Response: We thank the referee for these positive remarks and all excellent comments below.

However, at this stage, it is unclear whether the authors' findings overlap with previous work on cell competition in the pre- and early post-implantation mouse embryo (something that should be at least discussed).

Response: We agree with the referee that the concept of cell competition in the mouse embryo is very interesting and so far none of the published work, to our knowledge, examined cell competition in the context of mosaic aneuploidy in the pre-implantation embryo. The recent elegant paper from Tristan Rodriguez's laboratory, examines the fate of tetraploid cells, which can have very different impacts on cell physiology than the aneuploidy induced in our model system. Tetraploidy quadruples the entire genome without effecting the relative stoichiometry of proteins and it can therefore impose different stresses on a cellular system compared to the proteotoxic stress, followed by upregulation of autophagy, faced by the reversine-treated embryos. We discuss this now in the revised manuscript. We say 'Recently it was shown that tetraploid embryonic stem cells (ESCs) undergo apoptotic elimination from mouse ESC-embryo chimeras via mTOR, when surrounded by diploid cells²⁰. However, tetraploidy and single-chromosome aneuploidy could exhibit different physiological responses in a cell since tetraploidy quadruples the entire genome without affecting the relative stoichiometry of proteins, and therefore similar mechanisms might not apply to cell elimination in the model that we present here'. (page 12)

In addition, I thought that, although most of the major conclusions made by the authors were supported by the data, there were specific places in the manuscript where the data was not entirely convincing and could be improved upon.

Please find below a list of points that I would like to put forward as I think they would strengthen the manuscript if addressed.

1. Results with proportion of labelled / non-labelled cells

There is a certain proportion of diploid-aneuploid chimeras that end up with 100% of non-fluorescent cells coming from reversine-treated embryos (Figure S3 and S4). In some cases, the frequency is similar to diploid-diploid chimeras (Figure S4D). This seems counter to the idea that aneuploid cells are eliminated

from the embryos. Is this due to technical limitations related to the reversine treatment? Maybe some embryos do not see their proportion of aneuploid cells increase after reversine treatment?

Response: Although the frequency of diploid–diploid and diploid–aneuploid embryos with a certain % of non–fluorescent cells is similar in some cases, the overall frequency distribution curves for the % of non–fluorescent cells differs for the two chimeras. Both the proportion of chimeras with 100% non–fluorescent cells as well as the proportion of chimeras with 0% non–fluorescent cells are used as indicators of the level of elimination of non–fluorescent cells. This is the reason we have shown graphs in Figure S3C, S3D and S4B in which we have analysed the statistical significance of the differences in %non–fluorescent cells in the EPI and PE of diploid–diploid and diploid–aneuploid chimaeras.

In response to the questions raised regarding the instances of 100% non–fluorescent cells in diploid–aneuploid embryos, these could be attributed to the imperfect effect of reversine treatment, whereby some cells may remain diploid following reversine treatment (as we present in the Supplementary Fig. 1c), or these cells may be affected by more tolerable aneuploidies which fail to be eliminated until the early post–implantation stages of development.

Also, unless I am mistaken, there is does not seem to be any statistical analysis for the results presented in Figure S3E and F and S4D and E. It would be important to see if the distribution of frequencies is different between diploid–diploid and diploid–aneuploid chimeras.

Response: We apologise for the lack of clarity. We have shown statistical analysis for differences in % of non–fluorescent cells between diploid–diploid and diploid–aneuploid chimeras in the EPI and PE lineages within the same figure, i.e., Figure S3C for S3E, Figure S3D for S3F, Figure S4B for S4D and Figure S4C for S4E. We have restructured the figure legend to indicate this.

2. Embryo transfer experiments

The authors use embryo transfers as a more “in vivo” approach to complement ICM *in vitro* cultures. However, the embryos were only transferred for 12 hours and this was followed by a further 36 hours in culture. This makes it hard to talk about “in vivo” conditions and I think it would therefore be appropriate to slightly reword some of the conclusions such as “Taken together, these results indicate that aneuploid cells are preferentially eliminated during remodeling of the epiblast, both *in vitro* and *in vivo*”. Also, I do wonder why the authors decided to culture the embryos after transfer considering they could have let them develop longer *in vivo* in some cases. Since they had to grow the embryos *in vitro* for live imaging experiments, I guess it might have been to keep the same conditions across several experiments. However, it would be good if this could be clarified.

Response: We have recovered embryos shortly after their transfer, meaning the period of implantation development *per se* occurred *in vivo*, in order to follow the dynamics of cell behaviour, which is only possible if embryos develop *in vitro*. As the referee suggested, we have rephrased our text to explain this better and to mention that we did transfers for two reasons: 1. to allow *in vivo* implantation and 2. to retain the trophectoderm lineage of cells. We say ‘We next wanted to determine the fate of aneuploid cells in the mosaic embryos which were allowed to implant *in vivo* and in which TE is not removed.’ (page 4).

3. Live imaging experiments

These experiments are understandably difficult to undertake. However, I thought that the results presented by the authors in figure 2 were currently of relatively limited added value for the following reasons:
- From the data provided, it is not always clear that the cells shown are dying of apoptosis. For example, the cell shown in Figure 2A looks like it could be dividing and then move out of the plane.

Response: In response to this comment, we have added more z-planes for each time-point in Figure 2A and 2B.

In response to the comment regarding Figure 2A, the size of the green blebs remains much smaller than the size of a cell, indicating they represent apoptotic bodies rather than daughter cells resulting from cell division. As one moves further in time, such debris is eventually removed from the embryo. Careful analyses of such time lapse movies allow us to conclude that aneuploid cells are undergoing apoptosis. We now provide more information about this in the text and say 'Our live embryo imaging revealed several morphological features characteristic of apoptosis¹⁸ in the green fluorescent aneuploid cells. This included nuclear condensation, followed by the formation of the apoptotic bodies and subsequent removal of the cellular debris' (page 5).

- There is no n number for Figure 2C and the embryo shown looks unhealthy, suggesting that the culture or imaging conditions may not be optimal.

Response: We have mentioned n number in the figure legend (n = 12 embryos). We considered this embryo healthy because its epiblast undertook full morphogenesis – including lumenogenesis and re-organisation of the lineages leading to the formation of an egg cylinder with a cup shaped epiblast and abutting TE.

- There is no quantification. There also appears to be a significant number of green cells that do not commit apoptosis during the duration of the experiment

Response: We apologise for this confusion. All the quantifications relating to the fate of aneuploid cells and their elimination have been provided in Figure 1 (and corresponding Supplementary Figures) and in Figure 2, we just show some examples of apoptosis we observed – we have stated this in the figure legend.

The referee is correct, not all aneuploid cells are eliminated by the end of the culture – as shown by the quantification in Figure 1c and Figure 1f.

4. Relative proportion of cells in epiblast (Figure 3A-C)

As I understand, the absolute number of cells was counted in the epiblast of diploid-diploid and diploid-aneuploid chimeras and this number was divided by the average number of cells in the epiblast of diploid-diploid chimeras. I think it would be good to clarify the rationale behind this decision.

Response: We used relative proportions to determine if the epiblast size (cell number) of post-implantation diploid-aneuploid chimeras was affected with respect to diploid-diploid chimeras. We now explain this by saying “relative number of cells in the EPI were analysed for both types of chimeras (relative to the average of diploid-diploid chimeras) at the end of the peri-implantation culture to investigate the level of size regulation of diploid-aneuploids with respect to diploid-diploids.” (page 22)

5. The pH3 immunofluorescence in Figure 3D does not look like it has worked very well and it would be reassuring if the authors could show better examples of pH3-positive cells.

Response: As requested we have now added an additional example of pH3-positive cells in diploid-aneuploid chimera (Figure 3d and below), in which pH3 marks the dividing cell as can be identified from the DAPI staining. We also present another section which shows better pH3 staining.

Figure 3d 8-cell diploid(red)-aneuploid and diploid(red)-diploid chimeras were generated at the 8-cell stage. Immunosurgery was performed at the late blastocyst stage. The ICMs were embedded in Matrigel and cultured in IVC medium for 48 h. The percentage of the number of pH3-positive red fluorescent EPI cells of the total red fluorescent epiblast cells was analysed for each chimera, for both diploid-diploid and diploid-aneuploid ICMs. Scale bars, 20 μ m. Squares indicate the magnified regions, scale bars, 10 μ m. Student's t-test and $**p < 0.01$. Diploid-diploid $n = 13$ chimeras and diploid-aneuploid $n = 21$ chimeras. For all the graphs, all data are mean \pm s.e.m.

6. The immunofluorescence data for HSP70, LC3B and in particular p62 are unclear. HSP70 signal is strangely localised subapically in blastomeres, is that to be expected?

In regard to the apical localisation of HSP70, this has been demonstrated in the following studies:

- Ikari, A. *et al.* Up-regulation of Sodium-dependent Glucose Transporter by Interaction with Heat Shock Protein 70*. *J. Biol. Chem.* **277**, 33338-33343 (2002).
- Mashukova, A. *et al.* Rescue of atypical protein kinase C in epithelia by the cytoskeleton and Hsp70 family chaperones. *J. Cell Sci.* **122**, 2491-2503 (2009).

The p62 aggregates are worryingly also present in the blastocyst cavity. Some additional controls (using siRNA or blocking peptides for example) would help clarify what is signal from noise.

[Redacted]

Reviewers' Comments:

Reviewer #2:

Remarks to the Author:

This is a very well-done study, extremely important for the field. The authors have properly addressed all of my previous concerns and I strongly support its publication.

Reviewer #3:

None

Reviewer #4:

Remarks to the Author:

The authors have overall adequately addressed all the points I raised in my review by providing clarifications and further evidence when needed. I therefore remain positive and enthusiastic about the manuscript. signal from noise.

REVIEWERS' COMMENTS:

Reviewer #2 (Remarks to the Author and Author's Responses):

This is a very well-done study, extremely important for the field. The authors have properly addressed all of my previous concerns and I strongly support its publication.

Response: We are very grateful to the referee for carefully reading our manuscript and for the overall feedback and for these positive remarks.

Reviewer #4 (Remarks to the Author and Author's Responses):

The authors have overall adequately addressed all the points I raised in my review by providing clarifications and further evidence when needed. I therefore remain positive and enthusiastic about the manuscript.

Response: We are very grateful to the referee for carefully reading our manuscript and for the overall feedback and for these positive remarks.